# Distinct roles of nonmuscle myosin II isoforms for establishing tension and elasticity during cell morphodynamics

Kai Weißenbruch[1,2†*], Justin Grewe[3,4†], Marc Hippler[1,5], Magdalena Fladung[1], Moritz Tremmel[1], Kathrin Stricker[1], Ulrich Sebastian Schwarz[3,4*], Martin Bastmeyer[1,6*]

[1]Zoological Institute, Karlsruhe Institute of Technology (KIT), Karlsruhe, Germany; [2]Institute of Functional Interfaces (IFG), Karlsruhe Institute of Technology (KIT), Karlsruhe, Germany; [3]Institute for Theoretical Physics, University of Heidelberg, Heidelberg, Germany; [4]BioQuant-Center for Quantitative Biology, University of Heidelberg, Heidelberg, Germany; [5]Institute of Applied Physics, Karlsruhe Institute of Technology (KIT), Karlsruhe, Germany; [6]Institute for Biological and Chemical Systems - Biological Information Processing (IBCS-BIP), Karlsruhe Institute of Technology (KIT), Karlsruhe, Germany

**Abstract** Nonmuscle myosin II (NM II) is an integral part of essential cellular processes, including adhesion and migration. Mammalian cells express up to three isoforms termed NM IIA, B, and C. We used U2OS cells to create CRISPR/Cas9-based knockouts of all three isoforms and analyzed the phenotypes on homogenously coated surfaces, in collagen gels, and on micropatterned substrates. In contrast to homogenously coated surfaces, a structured environment supports a cellular phenotype with invaginated actin arcs even in the absence of NM IIA-induced contractility. A quantitative shape analysis of cells on micropatterns combined with a scale-bridging mathematical model reveals that NM IIA is essential to build up cellular tension during initial stages of force generation, while NM IIB is necessary to elastically stabilize NM IIA-generated tension. A dynamic cell stretch/release experiment in a three-dimensional scaffold confirms these conclusions and in addition reveals a novel role for NM IIC, namely the ability to establish tensional homeostasis.

**\*For correspondence:**
kai.weissenbruch@kit.edu (KW);
schwarz@thphys.uni-heidelberg.de (USS);
bastmeyer@kit.edu (MB)

[†]These authors contributed equally to this work

**Competing interests:** The authors declare that no competing interests exist.

## Introduction

The morphodynamics of nonmuscle cells are strongly determined by the contractile actomyosin cytoskeleton, consisting of actin filaments and motor proteins of the nonmuscle myosin II (NM II) class (*Burnette et al., 2014*; *Chen et al., 2010*; *Gumbiner, 1996*; *Ingber, 2003*; *Vicente-Manzanares et al., 2009*). Individual NM II hexamers assemble into bipolar filaments of up to 30 hexamers with a typical size of 300 nm, termed myosin minifilaments. These minifilaments can generate tension between antiparallel actin filaments due to their ATP-dependent motor activity. NM II generated forces are transmitted throughout the cell by subcellular structures such as the actomyosin cortex and different stress fiber subtypes (SFs) (*Burnette et al., 2014*; *Hotulainen and Lappalainen, 2006*). Since SFs are anchored to the extracellular matrix (ECM) via integrin-based focal adhesions (FAs), adherent cells are able to sense the physical properties of their environment at the cell-substrate interface (*Geiger et al., 2009*), where high forces can be measured with traction force microscopy (*Balaban et al., 2001*; *Oakes et al., 2017*). In a reciprocal fashion, the cells adapt their actomyosin machinery and thereby remodel the cell shape during motion-dependent processes like cell spreading, cell division, or cell migration (*Fenix et al., 2016*; *Svitkina, 2018*; *Taneja et al., 2020*; *Vicente-Manzanares et al., 2009*; *Yamamoto et al., 2019*). Accordingly, actomyosin

contractility has to be continuously adapted to provide both, short-term dynamic flexibility and long-lasting stability (*Ingber, 2003*; *Mandriota et al., 2019*; *Matthews et al., 2006*).

To precisely tune the contractile output, mammalian cells contain up to three different types of myosin II hexamers, which possess different structural and biochemical features. All hexamer-isoforms, which are commonly termed NM IIA, NM IIB, and NM IIC, contain the same set of light chains but vary with respect to their heavy chains, which are encoded by three different genes (*Heissler and Manstein, 2013*). While the cell type-dependent expression, structure, and function of NM IIC is still not clear, the loss of NM IIA and NM IIB causes severe phenotypes in the corresponding KO-mice (*Conti et al., 2004*; *Ma et al., 2010*; *Takeda et al., 2003*; *Tullio et al., 1997*; *Tullio et al., 2001*; *Uren et al., 2000*). On the cellular level, the loss of NM IIA drastically impairs SF formation, FA elongation, and cellular force generation, while the loss of NM IIB only causes mild deficiencies during SF and FA consolidation, cell shape stabilization, and force generation (*Barua et al., 2014*; *Beach et al., 2017*; *Betapudi et al., 2006*; *Billington et al., 2013*; *Even-Ram et al., 2007*; *Jorrisch et al., 2013*; *Sandquist and Means, 2008*; *Sandquist et al., 2006*; *Shutova et al., 2012*; *Shutova et al., 2017*; *Vicente-Manzanares et al., 2008*; *Vicente-Manzanares et al., 2011*; *Vicente-Manzanares et al., 2007*). In addition, NM IIA and NM IIB are well characterized with respect to their structural and biochemical differences: NM IIA propels actin filaments $3.5\times$ faster than NM IIB and generates fast contractions (*Kovács et al., 2003*; *Wang et al., 2000*). NM IIB can bear more load due to its higher duty ratio (*Pato et al., 1996*; *Wang et al., 2003*). Cell culture studies furthermore revealed that NM IIA and NM IIB hexamers co-assemble into heterotypic minifilaments, with a NM IIA to NM IIB gradient from the front to the rear of the cell (*Beach et al., 2014*; *Shutova et al., 2014*). Recent publications provide detailed insights that the composition of these heterotypic minifilaments tune contractility during cell polarization and migration (*Shutova et al., 2017*) or cytokinesis (*Taneja et al., 2020*). Given such extensive cellular functions and the ubiquitous expression of the NM II isoforms, it is very important to decipher the interplay of the different NM II isoforms for cell shape dynamics and force generation.

Here, we address this challenge with a quantitative approach that combines cell experiments and mathematical modelling. Using CRISPR/Cas9 technology, we generated isoform-specific NM II-KO cells from the U2OS cell line, which is a model system for the investigation of SFs (*Hotulainen and Lappalainen, 2006*; *Jiu et al., 2017*; *Lee et al., 2018*; *Tojkander et al., 2015*; *Tojkander et al., 2011*). The phenotypes of NM IIA- and NM IIB-deficient cells have been extensively characterized on homogenously coated substrates. Here, we focus on structured substrates, which resemble more closely the physiological environment of mesenchymal tissues. We employ collagen gels (*Cukierman et al., 2002*), micropatterned substrates (*Lehnert et al., 2004*; *Bischofs et al., 2008*; *Kassianidou et al., 2019*; *Labouesse et al., 2015*; *Tabdanov et al., 2018*; *Théry et al., 2006*; *Zand and Albrecht-Buehler, 1989*), and 3D-printed scaffolds (*Brand et al., 2017*; *Hippler et al., 2020*). Our results show that – in contrast to a homogenously coated surface – a structured environment can support a cellular phenotype with invaginated actin arcs even in the absence of NM IIA-induced contractility. A quantitative cell shape analysis reveals significant differences between WT, NM IIA-KO, and NM IIB-KO cells. A scale-bridging mathematical model explains these differences based upon the different crossbridge cycling rates of the NM II isoforms. Our analysis suggests that in structured environments, the main role of NM IIA is to dynamically build-up tension that later is elastically stabilized by NM IIB, which is in agreement with reports on homogenous substrates (*Heuzé et al., 2019*; *Sandquist et al., 2006*; *Shutova et al., 2017*; *Taneja et al., 2020*). A cell stretching assay in 3D-printed scaffolds reveals that this complementary interplay is necessary to generate a stable and long-lasting force output when cells are under mechanical stress. While NM IIC does not seem to play any role for shape determination of single cells, the cell stretching assay reveals that it is required for establishing tensional homeostasis.

## Results

### CRISPR/Cas9-generated isoform-specific NM II-KO cells reveal the expected phenotypes on homogenously coated substrates

To validate the impact of the different NM II isoforms on the cellular phenotype, we used U2OS cells as standard model for the investigation of SFs (*Hotulainen and Lappalainen, 2006*; *Jiu et al., 2017*;

*Lee et al., 2018*; *Tojkander et al., 2015*; *Tojkander et al., 2011*) that expresses all three NM II iso-forms (*Figure 1—figure supplement 1A*). We used CRISPR/Cas9 to target the first coding exons of *MYH9*, *MYH10,* and *MYH14,* encoding for NMHC IIA, NMHC IIB, and NMHC IIC, respectively. The loss of protein expression was confirmed by western blot analysis and immunofluorescence (*Figure 1—figure supplement 1A and B*). Additionally, DNA sequence analysis revealed frameshifts and pre-mature stop codons in exon 2 of *MYH9* and *MYH10* (*Figure 1—figure supplement 1C and D*).

Analyzing the NM II-KO phenotypes on substrates homogenously coated with fibronectin (FN) revealed comparable results to previous reports, where NM II isoforms were depleted via RNAi (*Cai et al., 2006*; *Sandquist et al., 2006*; *Shutova et al., 2017*; *Thomas et al., 2015*; *Vicente-Manzanares et al., 2007*) or genetic ablation (*Bridgman et al., 2001*; *Conti et al., 2004*; *Even-Ram et al., 2007*; *Lo et al., 2004*; *Ma et al., 2010*; *Takeda et al., 2003*; *Tullio et al., 1997*).Polarized U2OS WT cells form numerous SFs of different subtypes, as previously described (*Hotulainen and Lappalainen, 2006*; *Figure 1A*). Depletion of NM IIA leads to a markedly altered cellular phenotype with a branched morphology and several lamellipodia (*Figure 1B*; *Doyle et al., 2012*; *Even-Ram et al., 2007*; *Sandquist et al., 2006*; *Shih and Yamada, 2010*; *Shutova et al., 2017*). No ordered SF-network is built up and only few ventral SFs remain. Mature, elongated FAs are absent in NM IIA-KO cells (*Figure 1E&F* and *Figure 1—figure supplement 2A and B*). The effect of the NM IIB-KO is less severe and does not affect overall cell morphology (*Figure 1C*). All subtypes of SFs are present, but their distinct cellular localization is missing in many cells. Numerous mature FAs were observed throughout the cell body but their frequency was lower compared to WT cells (*Figure 1E and F* and *Figure 1—figure supplement 2A and B*). The depletion of NM IIC did not reveal any phenotypic differences compared to WT cells (*Figure 1D–F* and *Figure 1—figure supplement 2A and B*). In addition, the cell area does not significantly differ between WT and NM II-KO cells (*Figure 1—figure supplement 2C*).

To investigate if the loss of a certain NM II isoform has an impact on the remaining NM II paralogs, we compared the localization and intensity of NM IIA-C in WT cells and the respective NM II-KO cell lines (*Figure 1—figure supplements 3E* and *4*). In polarized WT cells, NM IIA and NM IIC signals are uniformly distributed throughout the cell body whereas NM IIB signals are enriched in the cell center (*Figure 1—figure supplement 3A*), confirming previous findings (*Beach et al., 2014*; *Kolega, 1998*; *Shutova et al., 2012*; *Shutova et al., 2014*). Depleting NM IIA strongly alters the localization pattern of both remaining paralogs, NM IIB and NM IIC (*Figure 1—figure supplement 3B*). Only few NM IIB minifilaments cluster along the remaining SFs. The same trend was observed for NM IIC, where a large fraction of NM IIC minifilaments localize in the cell center. The intensities of NM IIB or NM IIC minifilaments are both slightly increased but not significantly different, when NM IIA is depleted (*Figure 1—figure supplement 3E* and *Figure 1—figure supplement 4*). In contrast, no altered localization of the remaining paralogs was observed in NM IIB-KO cells and only the intensity of NM IIC was slightly but not significantly increased (*Figure 1—figure supplement 3C and E* and *Figure 1—figure supplement 4*). No differences were observed for the localization and intensity of NM IIA or NM IIB in NM IIC-KO cells (*Figure 1—figure supplement 3D and E* and *Figure 1—figure supplement 4*). Thus, only the loss of NM IIA had an distinct impact on the paralog localization.

## Overexpression of NM IIB does not compensate for the loss of NM IIA

Several studies identified NM IIA as the most abundant expressed isoform, while NM IIB and NM IIC are less strongly expressed (*Beach et al., 2014*; *Bekker-Jensen et al., 2017*; *Betapudi et al., 2011*; *Ma et al., 2010*). Thus, the knockout of NM IIA causes not only the loss of the motors distinct kinetic properties but also a drastic reduction of the totally available NM II hexamers. To determine the ratio of NM IIA to NM IIB minifilaments that assemble in WT cells, we generated fluorescent knock-in cells (*Koch et al., 2018*), where GFP is expressed under the endogenous promoter of NM IIA or B (*Figure 2A*). Given the heterozygous expression of GFP-NM IIA and B, respectively (*Figure 2—figure supplement 1*), our measurements do not represent absolute numbers of molecules but rather a relative estimation of the ratio of NM IIA and NM IIB hexamers in minifilaments. Measuring GFP signals along segmented actin fibers revealed that the ratio of NM IIA to NM IIB is roughly 4.5:1 (*Figure 2A&B*).

To test whether larger amounts of NM IIB are able to phenocopy the WT situation in NM IIA-KO cells, we overexpressed GFP-NM IIB under a constitutive active promoter. The relative amount of

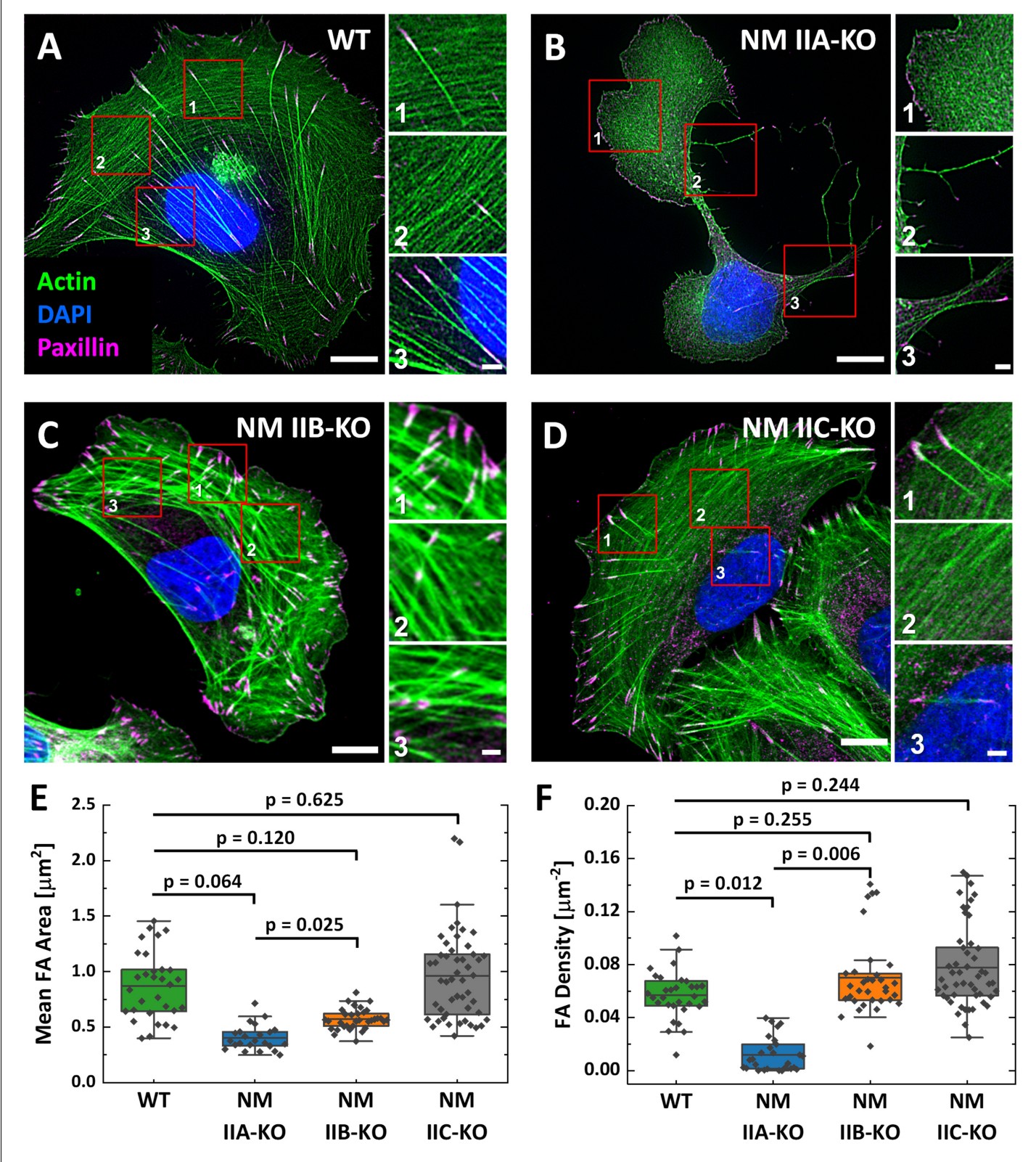

**Figure 1.** Phenotypes of NM IIA, NM IIB, and NM IIC-KO cell lines on homogenously coated substrates are very distinct. (**A**) U2OS WT cells show a polarized phenotype with prominent dorsal stress fibers (dSF) (1), transverse arcs (tA) (2), and ventral stress fibers (vSF) (3). Mature focal adhesions (FA) are visualized by elongated paxillin clusters that localize at the distal ends of dSF or both ends of vSF. (**B**) The NM IIA-KO leads to drastic morphological changes and the loss of most SFs and mature FAs. The overall actin structure resembles a dense meshwork of fine actin filaments (1). At
*Figure 1 continued on next page*

*Figure 1 continued*

the trailing edge, long cell extensions remain (2) and the only bundled actin fibers resemble vSF (3). (C) NM IIB-KO cells reveal slight changes in SF organization and FA structure. dSF (1), tA (2) and vSF (3) are present but their distinct localization pattern is disturbed. (D) The phenotype of NM IIC-KO cells is comparable to the WT. dSF (1), tA (2) and vSF (3) localize in a distinct pattern along the cell axis of polarized cells. (E) The mean FA area per cell is reduced for NM IIA-KO and NM IIB-KO cells, whereas FA density is only reduced in NM IIA-KO cells (F). Scale bars represent 10 µm for overviews and 2 µm for insets of (A) - (D).

The online version of this article includes the following figure supplement(s) for figure 1:

**Figure supplement 1.** Knockout of NMHC IIA, NMHC IIB and NMHC IIC via CRISPR/Cas9.

**Figure supplement 2.** Quantification of FA number and cell spreading area.

**Figure supplement 3.** Paralog localization in NM II-KO cells.

**Figure supplement 4.** Intensity quantification of NM II-paralogs.

overexpressed GFP-NM IIB is roughly 1.7-fold increased to endogenous GFP-NM IIA (*Figure 2A and B*) and in addition, these cells also express endogenous, unlabeled NM IIB. However, the distribution of NM IIB was still strongly clustered along single SF and does not compare to the distribution of NM IIA in WT cells (*Figure 2A*). We next used RLCs, phosphorylated at Ser19 (pRLC), as a isoform-independent marker staining for all active NM II molecules and compared the fluorescence intensities in WT-, NM IIA-KO-, and NM IIA-KO-cells overexpressing NM IIB-mApple. While the pRLC intensity was drastically reduced in NM IIA-KO cells, even strongest overexpression of NM IIB could not phenocopy the pRLC level of WT cells (*Figure 2C*). Similar results were achieved by comparing SF formation and FA maturation of GFP-NM IIB overexpressing NM IIA-KO cells to untransfected NM IIA-KO and WT cells (*Figure 2D and E*). Even high NM IIB levels did not induce the formation of an ordered actin cytoskeleton with bundled SFs of different subtypes. Similarly, FA maturation is still impaired when GFP-NM IIB is overexpressed (*Figure 2E*). In addition, we performed AFM nanoindentation experiments to compare the surface tension of WT and NM II-KO cells. While NM IIA-KO cells possess a significantly lower surface tension than WT cells, tension is significantly increased in NM IIB-KO cells (*Figure 2F*). No difference between WT and NM IIC-KO cells was observed. These results are in line with the interpretation that the different kinetic properties of NM IIA and NM IIB tune the contractile properties of the actin cortex and are supported by recently published data obtained by micropipette aspiration assays (*Taneja et al., 2020*). Thus, not only the absolute quantity of NM II molecules but rather the qualitative properties of both, NM IIA and B, are necessary for the formation of a fully functional actomyosin cytoskeleton.

## Structured environments reveal distinct functions of NM IIA and NM IIB in cell shape determination

To investigate the impact of the NM II-KOs in a more physiological inhomogenous and structured environment (*Ruprecht et al., 2017*), we next cultured U2OS WT and NM II-KO cell lines in a 3D collagen matrix (*Figure 3—figure supplement 1*). To probe the contractile properties of our NM II-KO cells, we cultivated cell seeded collagen gels (CSCGs) in suspension for 20 hr and measured the gel area at the beginning (red circled area) and the end of the experiment (blue area) (*Figure 3—figure supplement 1A*). We found that the gel contraction was highest in WT and NM IIC-KO gels, followed by slightly reduced values when using NM IIB-KO cells, and almost no contraction for NM IIA-KO cells. To connect these observations to the cellular morphologies of our NM II-KO cells, we next encased the cells in collagen gels that were attached to the coverslip (*Figure 3—figure supplement 1B–E*). All cell lines flattened in the collagen gel and in contrast to the cell morphologies on FN-coated coverslips, we now observed phenotypes with concave, inward bent actin arcs that line the cell contour as previously described for various cell types (*Bischofs et al., 2008*; *Brand et al., 2017*; *Kassianidou et al., 2019*; *Labouesse et al., 2015*; *Tabdanov et al., 2018*; *Théry et al., 2006*). This phenotype was most pronounced for WT, NM IIB-KO, and NM IIC-KO cells, while the phenotype of the NM IIA-KO cells showed many lamellipodial protrusions. However, these protrusions are also often intersected by small actin arcs.

Since a quantitative evaluation of these actin arcs was not feasible in the structurally ill-defined collagen matrix, we next changed to micropatterned substrates, which allowed us to normalize the cellular phenotypes. We produced cross-shaped FN-micropatterns via microcontact printing, which restrict FA formation to the pattern but still provide a sufficient adhesive area for the spreading of

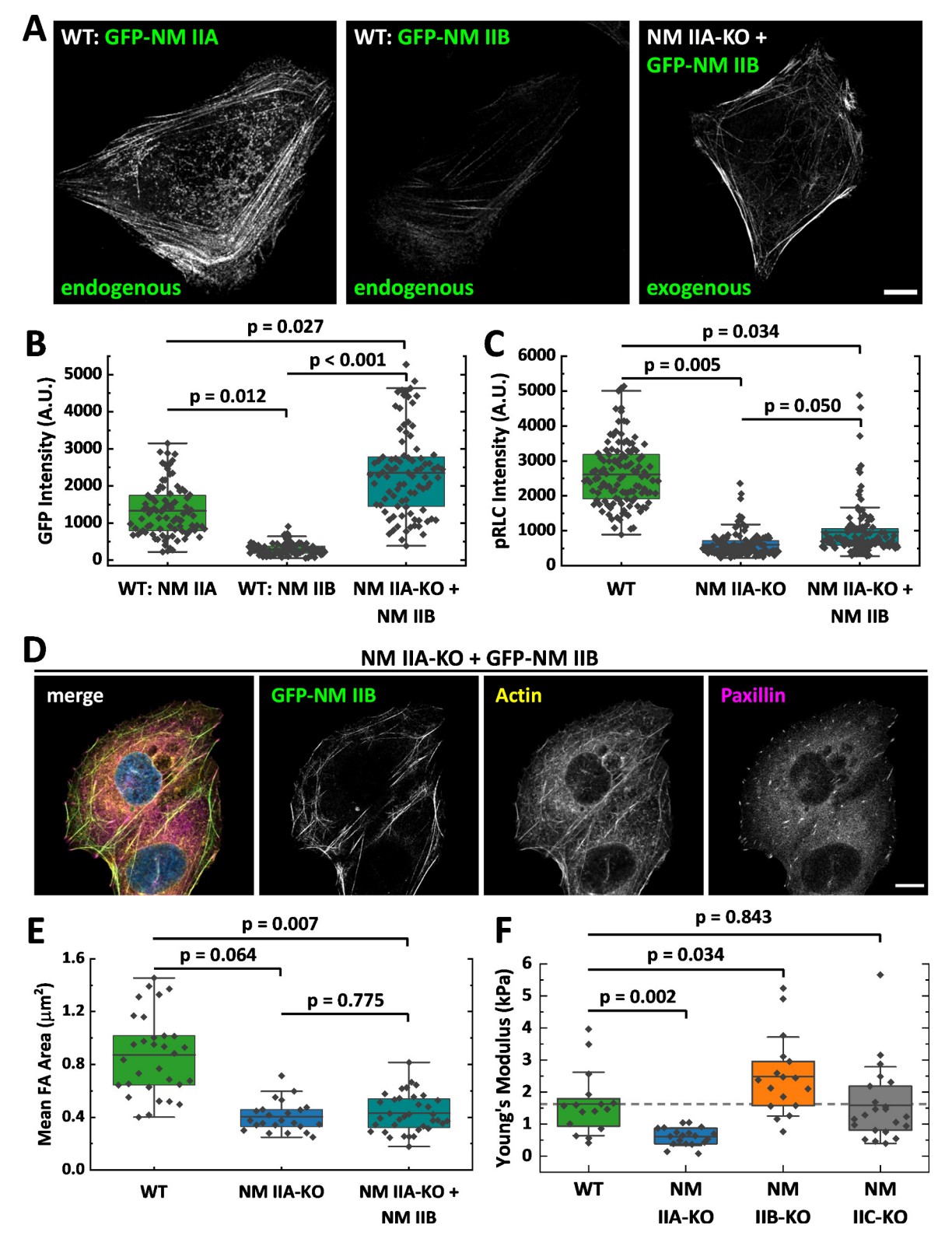

**Figure 2.** Overexpression of NM II B cannot compensate for the loss of NM II A. (**A**) GFP intensities were measured along segmented actin fibers to compare the ratios of NM IIA and NM IIB in WT and NM IIB overexpressing NM IIA-KO cells. When expressed under the endogenous promoter, the mean GFP intensity of NM IIB is 4.5 times lower compared to NM IIA. To increase the total amount of NM IIB without the interference of NM IIA, NM IIA-KO cells were transiently transfected with exogenous GFP-NM IIB under the CMV promoter. Even in NM IIA-KO cells with high NM IIB expression,

*Figure 2 continued on next page*

*Figure 2 continued*

NM IIB filament distribution is less homogenous compared to NM IIA in WT cells. (B) Quantitative comparison of endogenous NM IIA and NM IIB in WT cells, and exogenous NM IIB in NM IIA-KO cells. (C) PRLC signal intensity was used as a marker for active NM II filaments. The quantitative comparison of WT, NM IIA-KO, and NM IIB overexpressing NM IIA-KO cells shows a substantial reduction of pRLC intensity in NM IIA-KO cells, which is not restored in NM IIB overexpressing NM IIA-KO cells. (D) NM IIB overexpression does not phenocopy the WT situation. Immunfluorescent labeling of the actin cytoskeleton (yellow) and the FA marker paxillin (magenta) revealed that SFs are still sparse and FAs are less mature. (E) Quantitative comparison of the mean FA size in WT, NM IIA-KO, and NM IIB overexpressing NM IIA-KO cells. The values for WT and NM IIA-KO are the same as for *Figure 1E* and are only shown for comparison. (F) AFM nanoindentation experiments were performed to measure the surface tension of the NM II-KO cell lines. Compared to the WT, NM IIA-KO cells possess a significantly lower surface tension, while it is significantly higher for NM IIB-KO cells. No difference was observed for NM IIC-KO cells. Scale bar represents 10 µm in (A) and (D).

The online version of this article includes the following figure supplement(s) for figure 2:

**Figure supplement 1.** Generation of GFP-NM IIA and GFP-NM IIB fusion proteins.

U2OS cells (see Materials and methods section for details). Like in collagen gels, all cell lines adapted their shape to the pattern and gained a striking phenotype with concave, inward bent actin arcs that line the cell contour (*Figure 3*). In contrast to homogenously coated substrates, WT and all NM II-KO cell lines reveal a similar phenotype: Actin arcs bridge the passivated substrate areas and have a round shape, which we show to be very close to circular (*Figure 3—figure supplement 2*). NM II minifilaments localize along the circular actin arcs (*Figure 3—figure supplement 3B–D*), indicating that they are contractile SFs. From a geometrical point of view, the circularity results from two different NM II-based contributions to cell mechanics: tension in the cortex (surface tension σ) and tension in the actin arcs (line tension λ). Balancing these tensions can explain circular actin arcs with the radius $R = \lambda/\sigma$ (Laplace law). Typical order of magnitude values have been shown to be $R = 10$ µm, $\lambda = 20$ nN and $\sigma = 2$ nN/µm, with the values for λ and σ being extracted from for example traction force microscopy on soft elastic substrates or on pillar arrays (*Bischofs et al., 2009*). In addition, $R$ depends on the spanning distance $d$ between two adhesion sites, with larger $d$ leading to larger $R$ values (*Bischofs et al., 2008*). This dependence can be explained by assuming an elastic line tension $\lambda(d)$ (tension-elasticity model, TEM), suggesting that the mechanics of the peripheral SFs are not only determined by force generating NM II motors, but also by elastic crosslinking, for example by the actin crosslinker α-actinin.

To analyze potential differences in the NM II-KO cell lines, we measured arc radius $R$ and spanning distance $d$ and compared their correlation (*Figure 3A* insert; see methods section for details). WT cells regularly form actin arcs along all cell edges (*Figure 3A*). Both, NM IIA and NM IIB co-localize with the actin arcs (*Figure 3—figure supplement 3B*). Quantitative evaluation showed a positive correlation (r = 0.63 ± 0.06) of $R$ with increasing $d$, as observed previously (*Bischofs et al., 2008*; *Brand et al., 2017*; *Tabdanov et al., 2018*). Surprisingly, NM IIA-KO cells formed circular arcs and also obeyed a clear $R(d)$-correlation (r = 0.61 ± 0.06) (*Figure 3B*), despite the fact that their phenotype was strongly affected on homogenously coated FN-substrates. This agrees with our observations in the collagen gels and shows that the structured environment can support an invaginated phenotype even in the absence of NM IIA-induced contractility. In detail, however, we noticed marked differences compared to WT cells. Although actin arcs along the cell edges are still visible, they do not form as regular as in WT cells. The cell body often covers smaller passivated substrate areas but rather spreads along the crossbars, leading to smaller arcs. Compared to WT cells, fewer internal SFs are present as shown by a weaker image coherency (*Figure 3—figure supplement 3A*). NM IIB minifilaments co-localize along the actin arcs in NM IIA-KO cells and the pRLC staining is almost completely absent, suggesting that contractile forces are low in these cells (*Figure 3—figure supplement 3C*). Surprisingly, in NM IIB-KO cells the $R(d)$-correlation was strongly reduced (r = 0.33 ± 0.09) (*Figure 3C*), caused by the presence of a mixed population of bent and almost straight arcs that develop independent of the spanning distance $d$. Along these arcs, staining for NM IIA minifilaments and pRLC was comparable to WT cells (*Figure 3—figure supplement 3D*). We also quantified the degree of internal SF formation but did not find a difference between WT and NM IIB-KO cells (*Figure 3—figure supplement 3A*). NM IIC-KO cells did not reveal any differences concerning their morphology and the $R(d)$-correlation was comparable to WT cells (r = 0.63 ± 0.06) (*Figure 3D*). Importantly, overexpressing GFP-NM IIB in NM IIA-KO cells did not restore the WT phenotype (*Figure 3—figure supplement 4A*). These cells still spread along the cross bars and do not span over

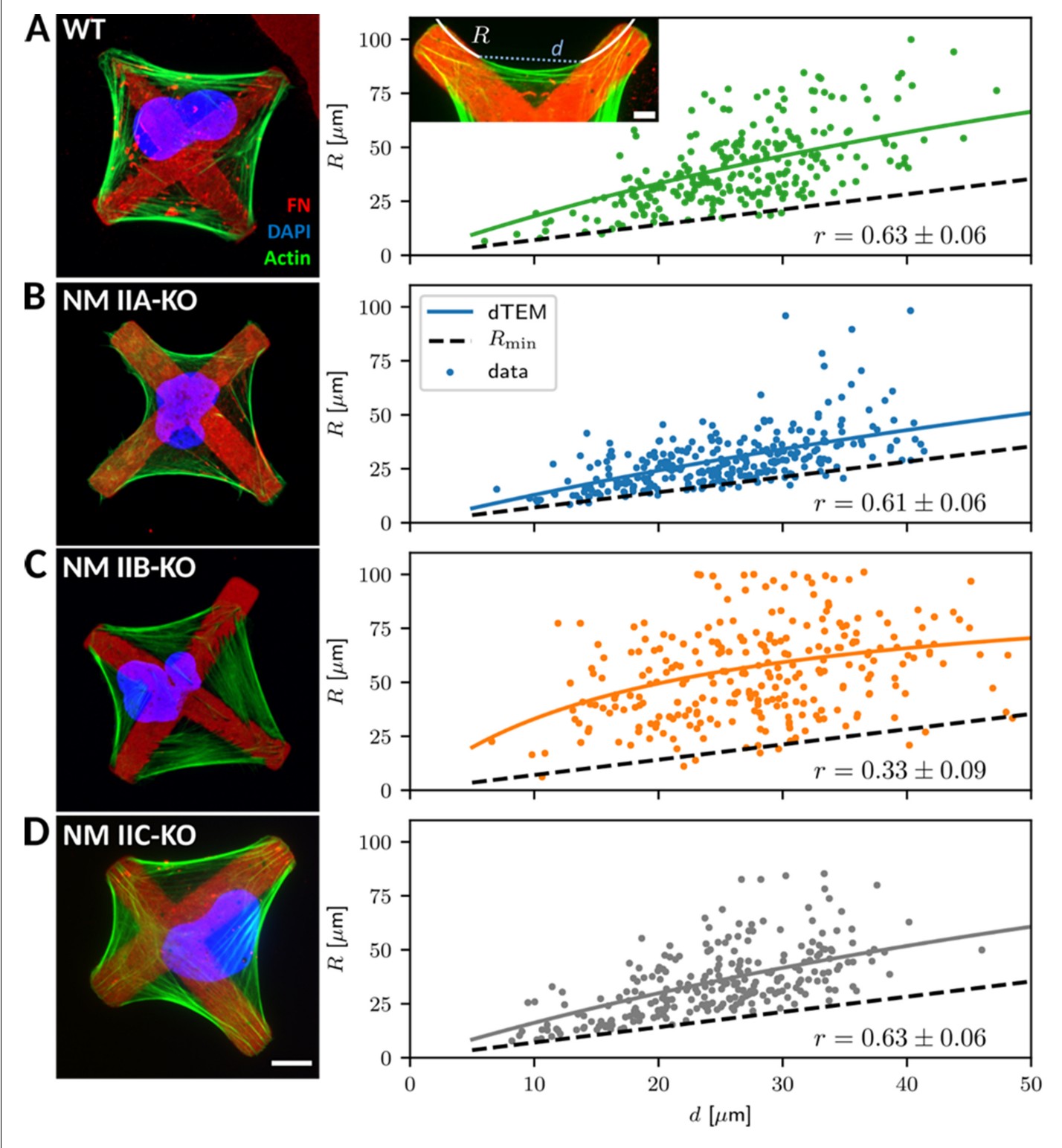

**Figure 3.** Quantitative shape analysis on cross-shaped micropatterns reveals distinct phenotypes for NM IIA-KO and NM IIB-KO cells. (**A**) U2OS WT cells show prominent invaginated actin arcs along the cell contour, with an invagination radius $R$ and a spanning distance $d$ (see inset). Quantitative image analysis reveals a positive $R(d)$-correlation (correlation coefficient r given at bottom right). Solid lines denote the bootstrapped mean fit of the dynamic tension-elasticity model (dTEM), black dashed lines denote the geometrically possible minimal radius. (**B**) U2OS NM IIA-KO cells form invaginated shapes on the cross-patterns despite their strongly perturbed shapes on homogenous substrates. The spanning distance of the arcs is shorter, but the positive correlation between $R$ and $d$ remains. (**C**) Actin arcs of U2OS NM IIB-KO cells are less invaginated compared to WT cells and

*Figure 3 continued on next page*

*Figure 3 continued*

their measured *R*(*d*)-correlation is very weak. (**D**) The phenotype of NM IIC-KO cells is comparable to WT and the *R*(*d*)-correlation is not affected. Scale bar represents 10 µm for (**A**) – (**D**).

The online version of this article includes the following figure supplement(s) for figure 3:

**Figure supplement 1.** NM II-KO phenotypes in collagen gels show distinct contractile behaviors with prominent actin arcs along the cell contour.

**Figure supplement 2.** Circularity of invaginated actin arcs.

**Figure supplement 3.** Localization of NM IIA and NM IIB minifilaments along peripheral actin arcs.

**Figure supplement 4.** NM IIA or B overexpression on micropatterned substrates show the importance of the different motor qualities.

---

large passivated areas. The *R*(*d*)-correlation (r = 0.63 ± 0.06) was comparable to WT and NM IIA-KO cells. Similarly, overexpressing GFP-NM IIA in NM IIB-KO cells did not significantly increase the *R*(*d*)-correlation (r = 0.49 ± 0.07) (**Figure 3—figure supplement 4B**). These results reveal opposing functions for NM IIA and NM IIB in cell shape determination on structured substrates. NM IIA-KO cells form actin arcs with small arc radii that are correlated to the spanning distance, while NM IIB-KO cells form actin arcs with large arc radii that are not correlated to the spanning distance.

## NM IIA and NM IIB contribute to dynamic generation of tension and elastic stability, respectively

To better understand these experimental results, we used mathematical models that connect our experimental findings to force generation by NM II motors. The main difference in the crossbridge cycles are that NM IIA generates faster contractions (**Wang et al., 2000**; **Kovács et al., 2003**) and NM IIB bears more load (**Wang et al., 2003**; **Pato et al., 1996**). We developed a dynamical tension-elasticity model (dTEM) that connects the stationary cell shapes to the dynamic crossbridge cycling. Due to geometrical constraints, the circular arcs on our cross-shaped micropattern can have central angles of only up to 90°, which defines a minimal radius $R_{min} = d/\sqrt{2}$ possible for a given spanning distance *d* (**Figure 4A**). We consider the SF as a dynamic contractile structure that sustains a continuous transport of cytoskeletal material from the FA towards the center of the SF (**Figure 4B**). This flow can be observed experimentally in ventral SFs for cells on homogenously coated substrates and in peripheral arcs for cells on cross-shaped micropatterns (**Figure 4—figure supplement 1** and **Figure 4—animations 1** and **2**) and, like retrograde flow, is believed to be driven by both actin polymerization in the FAs and myosin-dependent contractile forces (**Endlich et al., 2007**; **Shutova et al., 2017**; **Oakes et al., 2017**; **Russell et al., 2011**; **Tojkander et al., 2015**). Therefore, it should also depend on the isoform specific motor properties that result from the differences in the crossbridge cycles. Like in muscle cells, mature SFs are organized with sarcomeric arrangements of the myosin motors (**Dasbiswas et al., 2018**; **Shutova et al., 2017**; **Hu et al., 2017**). Accordingly, the number of serially arranged myosin motors increases linearly with SF length, and SF contraction speed should also increase with length. The stall force $F_s$, however, should not depend on the SF length because in this one-dimensional serial arrangement of motors, each motor feels the same force (**Thoresen et al., 2013**). Using an established model for the crossbridge cycle (**Figure 4C**) and the known differences between the powerstroke rates of NM IIA and NM IIB, we calculate the stall force $F_s$ for homotypic and heterotypic minifilaments (**Grewe and Schwarz, 2020a**; **Grewe and Schwarz, 2020b**). With increasing NM IIB content, the stall force increases and the free velocity decreases, which is mainly an effect of the much smaller duty ratio of NM IIA (**Appendix 1—figure 1**). For the polymerization at FAs, we assume that its rate increases with force, as has been shown in vitro for mDia1, the main actin polymerization factor in FAs (**Jégou et al., 2013**). Combining these molecular elements with the geometrical considerations of the TEM (details are given in Appendix 1), we arrive at a surprisingly simple form for the *R*(*d*) relation:

$$R(d) = \frac{d}{d_m + d} R_{max} \tag{1}$$

The maximal radius $R_{max} = F_s/\sigma$ is given by the ratio of stall force $F_s$ and surface tension σ. It can be understood as the arc radius that would be observed if there was no reduction of the tension by the inflow from the FAs and corresponds to a static TEM, with the stall force $F_s$ taking the role of the line tension λ in the Laplace law. As a NM II-KO is expected to affect both $F_s$ and σ to a similar

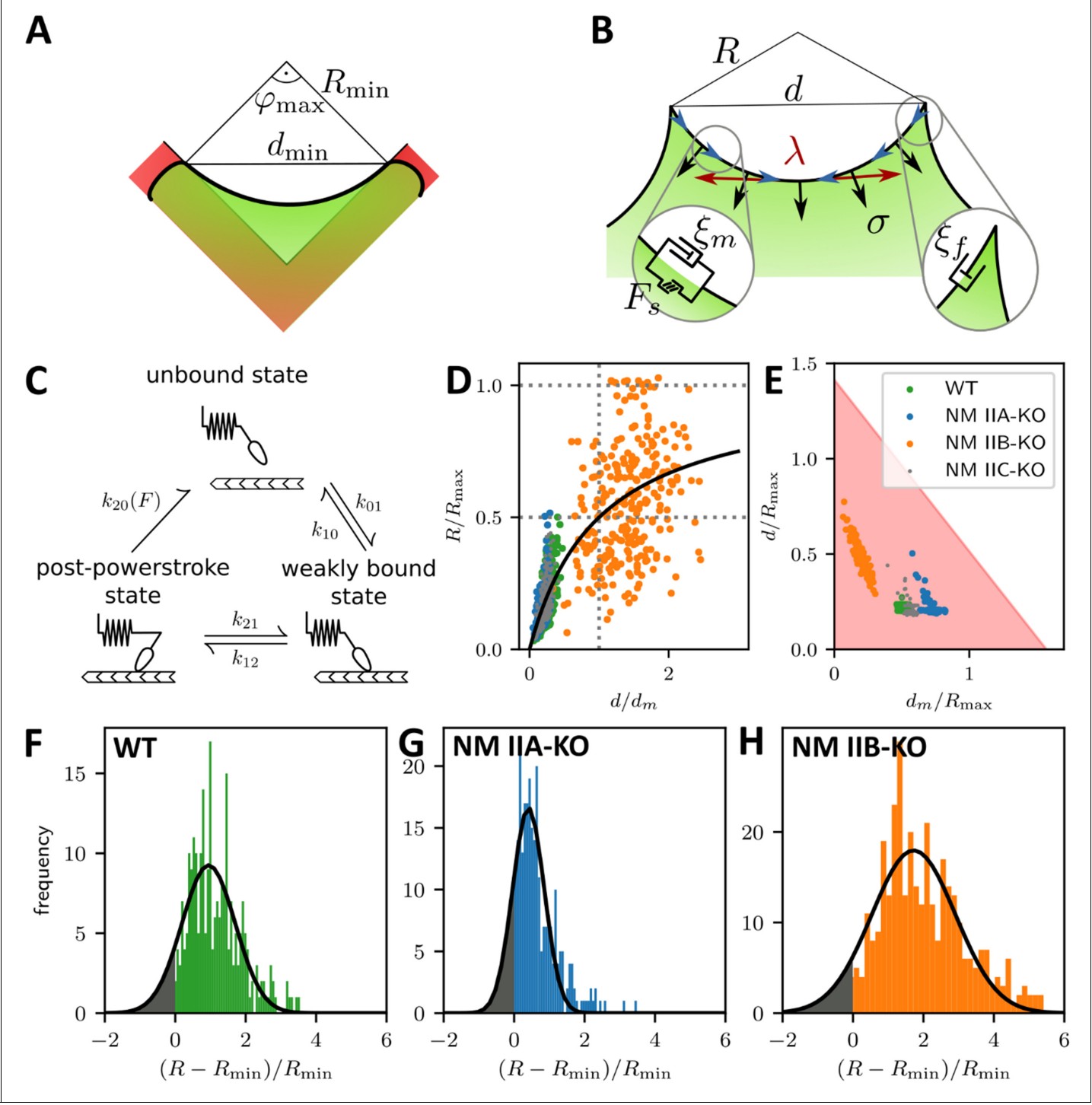

**Figure 4.** A dynamic tension-elasticity model (dTEM) connects the cellular phenotype to differences in the crossbridge cycling rates. (**A**) Illustration depicting the geometrically possible minimal radius on the cross-shaped micropattern. The circular arcs on our cross pattern can have central angles of only up to 90°. (**B**) Schematics of the mathematical model. At each point along the cell contour, line tension $\lambda$ and surface tension $\sigma$ balance each other and thereby determine the circular arc shape. The insets show the frictional elements required to allow flow of the peripheral fiber (friction coefficients $\xi_m$ and $\xi_f$ for stress fibers and focal adhesions, respectively). The motor stall force is denoted as $F_s$. (**C**) Illustration depicting the three main mechanochemical states during the crossbridge cycle and the corresponding model rates. (**D**) Normalizing experimental results using the fit parameters from (**A, B, C, D, E-H**) yields a master curve. WT, NM IIA-KO, and NM IIC-KO cells fall into the linear regime, NM IIB-KO cells into the plateau regime, which corresponds to a loss of correlation. (**E**) $d_m/R_{max}$ vs ratio of the maximum of the observed $d$-values to $R_{max}$. The region marked in red shows where the central angle of the arc is smaller than 90°. Points denote bootstrapped fit results. (**F–H**) Distributions of differences between observed radius

*Figure 4 continued on next page*

*Figure 4 continued*

and minimum allowed radius normalized to the minimally allowed radius resemble Gaussian distributions with cut-offs. From this, we can estimate the fraction of non-formed rods (gray areas).

The online version of this article includes the following figure supplement(s) for figure 4:

**Figure supplement 1.** Cytoskeletal flow in actin SFs of cells on homogenously coated substrates and in actin arcs of cells on cross-shaped micropatterns.

**Figure 4—animation 1.** GFP-NM IIA expressing U2OS WT cell showing cytoskeletal flow along SFs on homogenously-coated FN-substrates.

https://elifesciences.org/articles/71888#fig4video1

**Figure 4—animation 2.** GFP-NM IIA expressing U2OS WT cell showing cytoskeletal flow along actin arcs on cross-shaped FN-patterned substrates.

https://elifesciences.org/articles/71888#fig4video2

extent, our theory cannot predict directly how $R_{max}$ changes due to the loss of NM II. However, it predicts two regimes separated by the spanning distance at half maximal radius $d_m$, which is determined by the friction coefficients (Appendix 1): a linear regime at low and a plateau regime at high spanning distance, respectively.

Fitting *Equation 1* to the experimental data shown in *Figure 3A-D* yield the parameters $R_{max}$ and $d_m$ for each cell line (solid lines, dashed lines show the minimum radius resulting for a central angle of 90°). The mean fit values and standard deviations for the invaginated arcs are calculated from bootstraps and are listed in *Table 1*. By rescaling the experimental values using the fit parameters, the data points roughly follow a master curve (*Figure 4D*). For NM IIA-KO, we see that the data is in the linear regime with large $d_m$ values, corresponding to the high motor friction known for NM IIB (large duty ratio). This suggests that the measured radii for NM IIA-KO are smaller because the $R_{max}$ cannot be realized given the flow out of the FAs. We conclude that the main function of NM IIA is to dynamically generate tension. For NM IIB-KO, *Figure 4D* shows that the data is closer to the plateau regime. The small values for $d_m$ measured here corresponds to the low motor friction known for NM IIA (small duty ratio). This suggests that the measured radii for NM IIB-KO tend to be larger because the system can in fact dynamically sample $R_{max}$. At the same time, the independence of spanning distance $d$ also reflects the breakdown of the correlation, suggesting that NM IIB is required to elastically stabilize the arcs.

To further separate the different phenotypes, we plot our data in the two-dimensional parameter space of $(d_m/R_{max}, d/R_{max})$ (*Figure 4E*). The shaded region denotes allowed values due to the central angle being smaller than 90°. Strikingly, the ratio $d_m/R_{max}$, increases with the relative amount of NM IIB in the SF, from NM IIB-KO, over WT and NM IIC-KO cells to NM IIA-KO cells. This agrees with our theoretical finding that NM IIB stabilizes tension due to its slower crossbridge cycle (*Grewe and Schwarz, 2020a*; *Grewe and Schwarz, 2020b*).

Our results for the NM IIA-KO cells in *Figure 4E* are closest to the edge of the region with the theoretically permissible arcs, which suggests that in general some arcs cannot form because of geometrical constraints. *Figure 4F–H* show that the distribution of the difference of observed radius to the minimum radius, normalized to the minimum radius approximately follows a Gaussian distribution that is, however, cut off at zero difference. Assuming that the missing part of the distribution corresponds to the fraction of arcs that have not formed, we find that there should be approximately 10%, 18%, and 7% non-formed arcs for WT, NM IIA-KO and NM IIB-KO cells, respectively. This again

**Table 1.** Mean bootstrapped fit values for the invaginated arcs.

| Cell line | $R_{max}$ [µm] | $d_m$ [µm] | $d_m$ /$R_{max}$ |
|---|---|---|---|
| WT | 199 ± 5 | 100 ± 5 | 0.50 ± 0.02 |
| NM IIA-KO | 190 ± 24 | 137 ± 22 | 0.72 ± 0.04 |
| NM IIB-KO | 98 ± 18 | 20 ± 9 | 0.19 ± 0.05 |
| NM IIC-KO | 194 ± 19 | 110 ± 14 | 0.56 ± 0.03 |

The error is given as the standard deviation of the bootstrapped fit results. Note that the fit was bounded such that $R_{max}$ < 200 µm. Therefore, in cases where $R_{max}$ is close to 200 µm, the fit only gives reasonable error estimates for the quotient $d_m$ /$R_{max}$.

suggests that NM IIA is the most important isoform for the formation of arcs, while NM IIB is more important for stabilization.

## NM IIA-induced tension and NM IIB-derived elastic stability cooperate in the contractile output of cellular stress responses, while NM IIC mediates tensional homeostasis

We next investigated how the loss of NM II isoforms affects cellular contraction forces and the mechanoresponse upon extrinsic stretches (*Figure 5*). We applied our recently established method for the mechanical stimulation of single cells (*Hippler et al., 2020*). In brief, 3D micro-scaffolds composed of four non-adhesive walls, each with an inward directed protein-adhesive bar to guide cell attachment (schematically depicted in *Figure 5C*) were used to measure initial contraction forces. Cells cultured in these scaffolds attach to the bars, thus forming a cross-shaped morphology and pull the walls inwards. These movements were traced with time lapse microscopy for at least 1 hr, before the cells were detached by trypsinization (*Figure 5A*). Comparable to traction force microscopy on 2D substrates (*Balaban et al., 2001*), the displacement values were used to calculate the traction forces exerted by the cells and are in the following given as the sum of all four bars for each cell (*Hippler et al., 2020*). U2OS WT cells generated a mean initial force of 94 nN (*Figure 5B* and *Figure 5—animation 1*), while NM IIA-KO cells generated almost no traction forces (*Figure 5—animation 2* and *Figure 5—figure supplement 1B*) with a mean value of 11 nN (*Figure 5B*). When measuring the initial forces of NM IIB-KO cells, we obtained a mean value of 112 nN (*Figure 5B*, *Figure 5—animation 3* and *Figure 5—figure supplement 1C*), which did not significantly differ from WT cells. NM IIC-KO cells also showed no significant difference to the WT (*Figure 5—animation 4* and *Figure 5—figure supplement 1D*) with a mean force value of 110 nN (*Figure 5B*).

To investigate the mechanoresponse of single cells upon extrinsic applied forces, the above described 3D micro-scaffolds were complemented with a block of a guest-host based hydrogel (*Hippler et al., 2020*). Upon a chemical stimulus this hydrogel expands and pushes the walls apart, thus equibiaxially stretching the cell. Since the process is reversible, removing the stimulus causes a release of extracellular forces and a relaxation of the cell (*Figure 5C*). We applied the following workflow: WT or MN II KO cells were cultivated for 2 hr in the scaffolds to allow for attachment and equilibration of the cells. Then cells were imaged for 10–15 min before an equibiaxial stretch of ~ 20% was applied and the response (displacement in μm) was monitored for 30–60 min. After that time, the stretch was released and the cellular response again monitored for 20–40 min (*Figure 5D–G*). As previously described (*Hippler et al., 2020*), WT U2OS cells show a typical response to this stretch-release cycle: Cells counteract the stretch by increasing their traction forces over a time course of ~ 30 min until they settle on a new plateau value (*Figure 5—animation 5* and *Figure 5D*). After releasing the stretch, WT cells decrease their traction forces again and settle at the initial force value. When applying the stretch-release cycle to NM IIA-KO cells, the cells show no reaction, even after increasing the stretch period to 70 min (*Hippler et al., 2020*; *Figure 5E* and *Figure 5—animation 6*). In contrast, NM IIB-KO cells revealed a clear response to the stretch-release cycle similar to WT cells (*Figure 5F* and *Figure 5—animation 7*). NM IIC-KO cells also increased their traction forces upon stretching and the values were comparable to WT cells. However, 80% of the NM IIC-KO cells did not decrease their intracellular forces after release within the monitored timeframe of 30 min (*Figure 5G and H*, *Figure 5—animation 8* and *Figure 5—figure supplement 2D*).

As previously described (*Hippler et al., 2020*), displacements of the scaffolds can be transformed into cellular forces (*Figure 5H*). Quantifying the force increase (within the blue boxes in *Figure 5D–G*) showed a mean value of 72 nN for WT cells and no mean force increase for NM IIA-KO cells (*Hippler et al., 2020*). NM IIB-KO cells also revealed a force increase of 41 nN which was, however, significantly lower than in WT cells (*Figure 5H*). The force increase of NM IIC-KO cells did not significantly differ from WT cells, but the mean variation was higher. These data strongly support our hypothesis that NM IIA initiates cellular tension, while NM IIB contributes to this tension by providing elastic stability for a stable force output on longer time-scales, that is during the mechanoresponse of cells. In addition, NM IIC seems to play a role in the temporal control of the force relaxation, that is the mechanical homeostasis of cells.

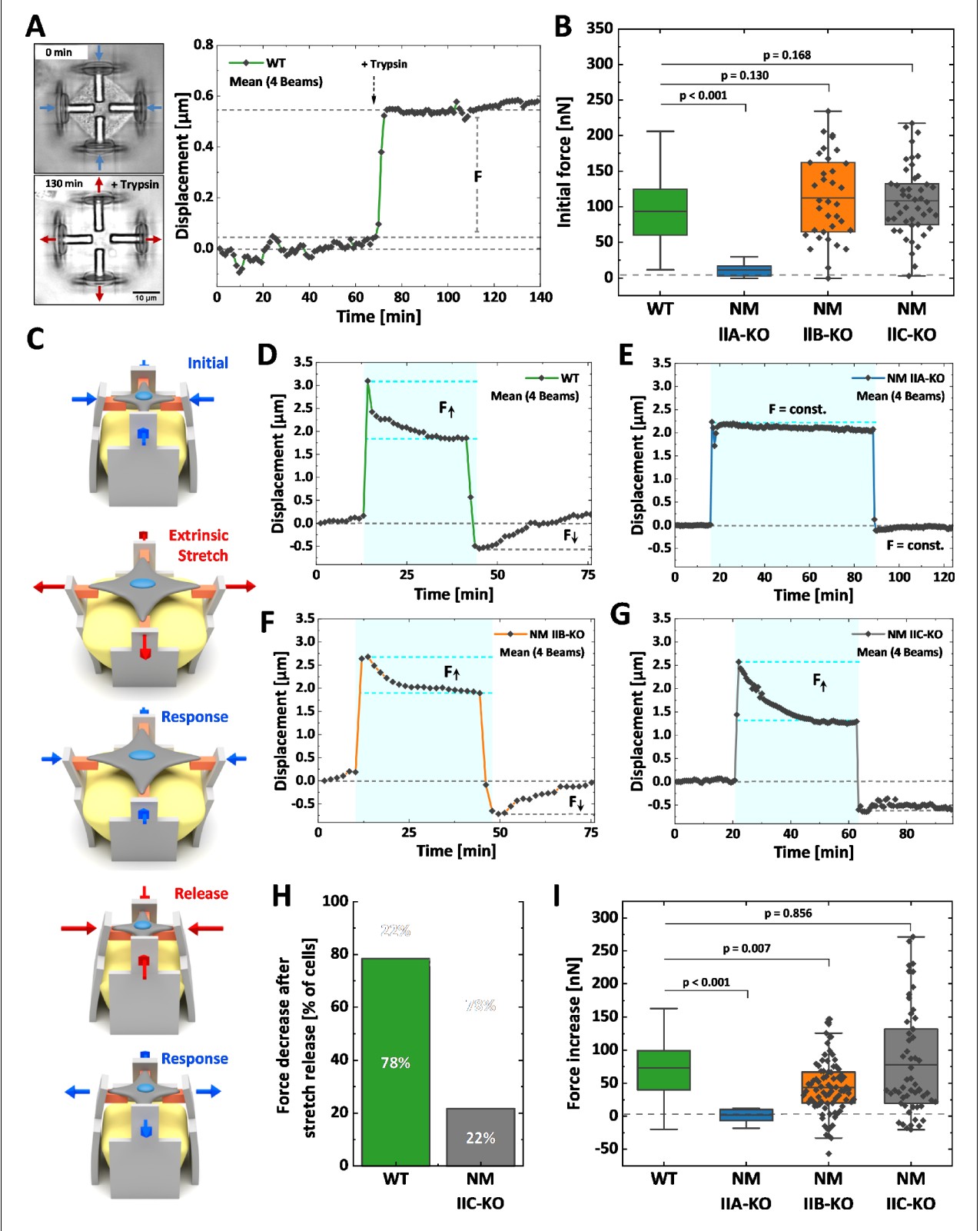

**Figure 5.** Differential contributions of NM II isoforms to cellular mechanoresponse. Cells were cultured 3D micro-scaffolds composed of four non-adhesive walls, each with an inward directed protein-adhesive bar to guide cell attachment. Cells attach to the bars, form a cross-shaped morphology and pull the walls inwards. (A) Initial cellular tractions forces were determined by detaching the cell from the scaffold using trypsin/EDTA and measuring the corresponding average beam displacement as indicated in the plot. (B) Comparison of the initial forces of the different cell lines. Data for WT and

*Figure 5 continued on next page*

*Figure 5 continued*

NM IIA-KO have been reproduced from (B) of *Hippler et al., 2020*, therefore only the mean values are shown (originally published under the Creative Commons Attribution-Non Commercial 4.0 International Public License (CC BY-NC 4.0; https://creativecommons.org/licenses/by-nc/4.0/. Further reproduction of this panel would need to comply with the terms of this license)). No significant differences were observed between WT (mean value = 94 nN), NM IIB-KO (mean value = 112 nN), and NM IIC-KO cells (mean value = 110 nN). A significant decrease was observed for NM IIA-KO cells (mean value = 11 nN). (C) Illustration depicting the stretch-release cycle applied to the cells. (D-G) Examples of average beam displacements (corresponding to *Figure 5—animations 5–8*) are plotted as a function of time. The blue area depicts the time frame, in which the corresponding cell was stretched. (D) WT cells actively counteract the stretch and increase their contractile forces until reaching a plateau after ~ 30–40 min. After releasing the stretch, cellular contraction forces remained high, but decreased to the initial level after 20–30 min. (E) No cellular force response is observed, when applying the stretch-release cycle to NM IIA-KO cells, even after longer stretch periods. (F) NM IIB-KO cells increase their force after stretching and reach a plateau after 30–40 min. The force increase is lower compared to WT cells. After releasing the stretch, NM IIB-KO cells also reduce their forces until the initial set point is reached. (G) NM IIC-KO cells increase their force upon the stretch but do not relax to the initial setpoint within the observed timeframe. (H) The quantification shows that a force decrease after the stretch release was observed for 78% of the WT cells but only for 22% of the NM IIC-KO cells. (I) Comparison of the force increase of WT and NM II-KO cells, after being mechanically stretched. Data for WT and NM IIA-KO reproduced from (D) of *Hippler et al., 2020*, therefore only the mean values are shown (originally published under the Creative Commons Attribution-Non Commercial 4.0 International Public License (CC BY-NC 4.0; https://creativecommons.org/licenses/by-nc/4.0/. Further reproduction of this panel would need to comply with the terms of this license)). A mean increase of 73 nN was observed for WT cells and no force increase for NM IIA-KO cells (mean value = 0.29 nN). Compared to WT cells, NM IIB-KO cells display a significantly lower force increase (mean value = 41 nN), while NM IIC-KO cells show a comparable mean value. However, higher variations in the force response are observed for NM IIC-KO cells. Scale bar represents 10 μm in (A). The online version of this article includes the following source data and figure supplement(s) for figure 5:

**Source code 1.** Matlab code for displacement tracking and force calculation.
**Figure supplement 1.** Initial force measurements of WT and NM II-KO cells.
**Figure supplement 2.** Displacements of individual beams for stretched WT and NM II-KO cells.
**Figure 5—animation 1.** Initial forces of a U2OS WT cell in a 3D-printed microscaffold.
https://elifesciences.org/articles/71888#fig5video1
**Figure 5—animation 2.** Initial forces of a U2OS NM IIA-KO cell in a 3D-printed microscaffold.
https://elifesciences.org/articles/71888#fig5video2
**Figure 5—animation 3.** Initial forces of a U2OS NM IIB-KO cell in a 3D-printed microscaffold.
https://elifesciences.org/articles/71888#fig5video3
**Figure 5—animation 4.** Initial forces of a U2OS NM IIC-KO cell in a 3D-printed microscaffold.
https://elifesciences.org/articles/71888#fig5video4
**Figure 5—animation 5.** Reactive forces of a U2OS WT cell in a 3D-printed microscaffold upon a stretch-release cycle.
https://elifesciences.org/articles/71888#fig5video5
**Figure 5—animation 6.** Reactive forces of a U2OS NM IIA-KO cell in a 3D-printed microscaffold upon a stretch-release cycle.
https://elifesciences.org/articles/71888#fig5video6
**Figure 5—animation 7.** Reactive forces of a U2OS NM IIB-KO cell in a 3D-printed microscaffold upon a stretch-release cycle.
https://elifesciences.org/articles/71888#fig5video7
**Figure 5—animation 8.** Reactive forces of a U2OS NM IIC-KO cell in a 3D-printed microscaffold upon a stretch-release cycle.
https://elifesciences.org/articles/71888#fig5video8

## Discussion

Here, we have systematically analyzed the roles of the three different NM II isoforms for cellular morphodynamics and force generation in structured environments with a combined experimental and theoretical approach. Using CRISPR/Cas9-technology and U2OS cells, we depleted for the first time all three isoforms from the same cellular background. Cell culture on homogenously coated substrates confirmed previous reports about NM IIA and NM IIB (*Even-Ram et al., 2007*; *Sandquist et al., 2006*; *Shutova et al., 2017*; *Vicente-Manzanares et al., 2011*; *Vicente-Manzanares et al., 2007*). Without NM IIA, cells lack global tension, SFs and mature FAs. The NM IIB-KO only leads to a less clear distinction between different types of SF and to smaller FAs. The amount of NM IIA in minifilaments is at least 4.5 times higher compared to its paralogs. However, we did not observe an upregulation of NM IIB or NM IIC upon the loss of NM IIA, and overexpressing NM IIB was not sufficient to compensate the observed effects, showing that the induced deficits are due to the loss of the distinct motor properties rather than the overall quantity of the NM II population.

To our surprise, the marked phenotypic differences of NM IIA-KO cells are dampened when the cells were cultivated in a structured environment. This suggests that NM IIA contributes in guiding

the cellular phenotype in the absence of external guiding cues. Although NM IIA-KO and NM IIB-KO cells superficially look similar on the micropattern, the quantitative cell shape analysis revealed marked differences. NM IIA-KO cells form small arcs that are correlated to the spanning distance and fail to bridge larger passivated substrate areas. NM IIB-KO cells form large actin arcs, which are poorly correlated with the spanning distance. Similar opposing effects were observed in AFM nano-indentation experiments: While the surface tension was reduced in NM IIA-KO cells, it was increased when NM IIB was depleted. Since the phenotypes on the micropattern arise from the interplay of the two types of actomyosin-mediated tension, σ and λ (*Bischofs et al., 2008*; *Brand et al., 2017*), we hypothesize that the different properties of NM IIA and NM IIB tune the contractile and mechanical properties of both types of tension in a similar manner. Although we cannot distinguish the absolute values of NM IIA or NM IIB that contribute to σ or λ, our results show that the loss of NM IIA reduces both values, while the loss of NM IIB leads to an increase of both types of tension.

By connecting the experimental results to our dynamic tension-elasticity model (dTEM), we can explain our results for the micropattern by differences in the molecular crossbridge cycles. NM IIA-KO cells still possess NM IIB-derived elastic stability but lack dynamic tension leading to low intracellular forces. Without NM IIA motors, NM IIB is too slow to rearrange the contractile forces in accordance with the fast polymerization of actin filaments at FAs. Consequently, the arcs on the crosspattern are smaller and more bent inwards, since the actin polymerization rate overpowers the motor stall force and the surface tension increases the curvature. Since the generation of contractile actomyosin bundles is a mechanosensitive process (*Tojkander et al., 2015*), a polarized actomyosin cytoskeleton is missing in NM IIA-KO cells. The only remaining SF resemble ventral SF, because their turnover is lowest (*Kumar et al., 2006*; *Lee et al., 2018*). NM IIB-KO cells in contrast still possess NM IIA minifilaments, which generate sufficient but unbalanced intracellular forces. The low motor stall force of NM IIA overpowers the actin polymerization rate leading to low curvatures of the actin arcs, which arise independent of the spanning distance. Although the fast and dynamic motor activity of NM IIA is sufficient to induce the mechanosensitive assembly of all SF subtypes, their distinct localization is disturbed (*Shutova et al., 2017*; *Vicente-Manzanares et al., 2008*). Likewise, loss of NM IIB does not affect the formation of FAs, however, they do not grow to full size, since the actin templates are not sufficiently stabilized by the cross-linking properties of NM IIB (*Vicente-Manzanares et al., 2011*). No phenotypic change or disturbance in the $R(d)$ correlation was observed when depleting NM IIC. Thus, our results indicate that NM IIC might be less important for the morphodynamics of single cells, at least in our cell line.

Our model focuses on the crossbridge cycle properties in the motor heads and is sufficient to explain the experimentally observed $R(d)$ relations on the micropattern. It is important to note that in addition, the NM II isoforms differ in their tail regions, leading to marked differences in minifilament assembly/disassembly dynamics and intracellular localization patterns (*Breckenridge et al., 2009*; *Juanes-Garcia et al., 2015*; *Kaufmann and Schwarz, 2020*). Future work has to address how the isoform-specific phosphorylation pattern of the c-terminal tails influence cell shape determination in structured environments, for example by using NM II chimeras or phospho-mutants (*Juanes-Garcia et al., 2015*; *Sandquist and Means, 2008*).

We confirmed our results in cell stretching experiments, where the cellular force generation during mechanical stress was precisely monitored for all three NM II-KO cell lines. Measuring cellular contraction forces in microstructured 3D scaffolds revealed a complete loss of forces for NM IIA-KO cells but no reduction for NM IIB-KO and NM IIC-KO cells. Upon a stretch-release cycle, WT cells show a behavior that was previously described as mechanical homeostasis (*Hippler et al., 2020*; *Webster et al., 2014*; *Weng et al., 2016*). Upon stretch, intracellular forces increase by a factor of two and equilibrated on this new setpoint. When releasing the stretch, cellular contraction forces remain high for a short period, but decrease to the initial level after 20–30 min. As expected, NM IIA-KO cells did not respond at all to the stretch-release cycle (*Hippler et al., 2020*). A clear response was observed when stretching NM IIB-KO cells, however, the force increase did not reach the values of WT cells. After releasing the stretch, cellular contraction forces also decreased to the initial level, however, force decrease was accompanied with oscillations of contractile pulses in about 50% of analyzed traces (*Figure 5—animation 7* and red trace in *Figure 5—figure supplement 2*) as compared to 10% in WT cells. Thus, we again conclude that NM IIB seems to regulate the spatio-temporal response of the actomyosin system by stabilizing NM IIA induced tension. We observed the same trend in collagen gels, where NM IIB-KO cells did contract the gel to a lower amount than

WT cells, which is in good agreement with results of others (*Meshel et al., 2005*). Unexpectedly, also NM IIC-KO cells behaved differently from the WT and displayed a delayed response after the stretch was released. In about 80% of the analyzed NM IIC-KO cells, contraction forces did not relax to the initial setpoint within the observed timeframe of 30 min, while this was only the case for 20% of the WT cells. Because the cellular function of NM IIC reported here seems not to be directly related to differences in the powerstroke cycle, future experiments might reveal, whether the loss of NM IIC leads to a delay in the cellular mechanoresponse by interfering with the global organization of the NM II contractome. Structural in vitro analysis revealed that NM IIC minifilaments are smaller compared to their paralog counterparts (*Billington et al., 2013*). As it was reported that NM IIA and NM IIC co-localize throughout the whole cell body in U2OS cells (*Beach et al., 2014*), this could suggest that NM IIC has a role as a scaffolding protein during the formation of higher ordered NM IIA minifilament stacks (*Fenix et al., 2016*), comparable to the role of myosin-18B (*Jiu et al., 2019*).In epithelial sheets, NM IIC was shown to regulate the geometry of the epithelial apical junctional-line (*Ebrahim et al., 2013*) and the microvilli length (*Chinowsky et al., 2020*). Strikingly, Beach and colleagues showed that the phenotypic switch during EMT (epithelial-mesenchymal transition) and the subsequent invasiveness of murine mammary gland cells goes along with a downregulation of NM IIC and an upregulation of NM IIB (*Beach et al., 2011*). Thus, NM IIC might be of special interest for the structural organization and integrity of epithelial cell sheets.

In summary, we showed that the environmental guidance of the actomyosin system follows a logical order. The initiation depends on the presence of NM IIA. This motor can quickly repopulate newly formed protrusions and initiate new contraction sites (*Baird et al., 2017*), giving rise to heterotypic minifilaments and dynamizing NM IIB (*Fenix et al., 2016*; *Shutova et al., 2017*). Once the contraction is initiated, NM IIB co-assembles into the preformed contraction site and stabilizes the tension, as this motor is prone to maintain tension on longer timescales by staying longer bound to the actin cytoskeleton (*Sandquist and Means, 2008*; *Vicente-Manzanares et al., 2008*). Thus, the stability of the heterotypic minifilaments is facilitated by the relative composition of NM IIA and NM IIB (*Kaufmann and Schwarz, 2020*). NM IIC might contribute to this set-up as a structural regulator that controls force maintenance and relaxation, especially in a dynamical context, when external conditions change and homeostasis has to be ensured. In a physiological context, such a self-assembling system would be able to precisely tune the contractile output of single cells but also cell collectives. Since different studies showed prominent functions of NM IIB and NM IIC during EMT and invasiveness (*Beach et al., 2011*; *Thomas et al., 2015*), or the reinforcement of cell-cell adhesion sites (*Heuzé et al., 2019*), our insights should be transferred to the tissue context, e.g. to explain collective migration effects in development, wound healing or cancer (*Scarpa and Mayor, 2016*; *Shellard and Mayor, 2019*; *Sunyer et al., 2016*; *Trepat and Sahai, 2018*).

# Materials and methods

## Key resources table

| Reagent type (species) or resource | Designation | Source or reference | Identifiers | Additional information |
|---|---|---|---|---|
| Gene (*Homo sapiens*) | MYH9 | NCBI | NC_000022.11 | |
| Gene (*Homo sapiens*) | MYH10 | NCBI | NC_000017.11 | |
| Gene (*Homo sapiens*) | MYH14 | NCBI | NC_000019.10 | |
| Cell line (*Homo sapiens*) | U2OS | ATCC | # HTB-96 RRID:CVCL_0042 | |
| Cell line (*Homo sapiens*) | U2OS NM IIA-KO | This paper | | CRISPR/Cas9-generated knockout cell line; compare Materials section 2 |
| Cell line (*Homo sapiens*) | U2OS NM IIB-KO | This paper | | CRISPR/Cas9-generated knockout cell line |

*Continued on next page*

*Continued*

| Reagent type (species) or resource | Designation | Source or reference | Identifiers | Additional information |
|---|---|---|---|---|
| Cell line (*Homo sapiens*) | U2OS NM IIC-KO | This paper | | CRISPR/Cas9-generated knockout cell line |
| Cell line (*Homo sapiens*) | U2OS GFP-NM IIA | This paper | | CRISPR/Cas9D10A-generated knock-in cell line; compare Materials section 2 |
| Cell line (*Homo sapiens*) | U2OS GFP-NM IIB | This paper | | CRISPR/Cas9-generated Knock-in cell line |
| Transfected construct (*Homo sapiens*) | CMV-GFP-NMHC IIA | RRID:addgene_11347 | PMID:11029059 | Gift from Robert Adelstein |
| Transfected construct (*Homo sapiens*) | CMV-GFP-NMHC IIB | RRID:addgene_11348 | PMID:11029059 | Gift from Robert Adelstein |
| Transfected construct (*Homo sapiens*) | mApple-MyosinIIB-N-18 | RRID:addgene_54931 | RRID:Addgene_54931 | Gift from Michael Davidson |
| Recombinant DNA reagent | pSPCas9(BB)—2A-Puro (PX459) V2.0 | RRID:addgene_62988 | PMID:24157548 | Gift from Feng Zhang |
| Recombinant DNA reagent | pX335-U6-Chimeric_BB-CBh-hSp Cas9n(D10A) | RRID:addgene_42335 | PMID:23287718 | Gift from Feng Zhang |
| Recombinant DNA reagent | pMK-RQ-*MYH9* | This paper | | Donor sequence for HDR, compare Materials section 2 and 5 |
| Recombinant DNA reagent | pMK-RQ-*MYH10* | This paper | | Donor sequence for HDR, compare Materials section 2 and 5 |
| Antibody | Anti-Alpha-Tubulin (mouse monoclonal) | Sigma-Aldrich | #T5168 RRID:AB_477579 | WB: (1:2000) |
| Antibody | Anti-fibronectin (mouse monoclonal) | BD Biosciences | #610077 RRID:AB_2105706 | IF: (1:500 250 µg/ml) |
| Antibody | Anti-NMHC IIA (rabbit polyclonal) | BioLegend | #909801 RRID:AB_2565100 | IF: (1:500) WB: (1:1000 1 mg/ml) |
| Antibody | Anti-NMHC IIB (rabbit polyclonal) | BioLegend | #909901 RRID:AB_2565101 | IF: (1:500) WB: (1:1000 1 mg/ml) |
| Antibody | Anti-NMHC IIC (D4A7) (rabbit monoclonal) | Cell signaling | #8189S RRID:AB_10886923 | IF: (1:100) WB: (1:1000) |
| Antibody | Anti-pMLC2 (Ser19) (mouse monoclonal) | Cell signaling | #3671S RRID:AB_330248 | IF: (1:200) |
| Antibody | Anti-Paxillin (mouse monoclonal) | BD Biosciences | #610619 RRID:AB_397951 | IF: (1:500 250 µg/ml) |
| Antibody | Anti-GFP (rabbit polyclonal) | abcam | #ab6556 RRID:AB_305564 | WB: (1:2000 0.5 mg/ml) |
| Peptide, recombinant protein | Fibronectin from human plasma | Sigma-Aldrich | #F1056 RRID:AB_2830099 | 10 µg/ml |

*Continued on next page*

*Continued*

| Reagent type (species) or resource | Designation | Source or reference | Identifiers | Additional information |
|---|---|---|---|---|
| Peptide, recombinant protein | Collagen I and Thin plate coating collagen I from rat tails | Enzo Life Sciences | #ALX-522–435 and #ALX-522-440-0050 | 1 mg/ml final concentration |
| Chemical compound, drug | 1-Adamantane carboxylic acid | Sigma-Aldrich | #106399 | 20 mM in DMEM pH 7 |
| Software, algorithm | Digital Image correlation and tracking Version 1.2.0.0 | MathWorks MATLAB (*Eberl et al., 2006*) | FileID 12413 | The custom-adapted script (see PMID:32967835) is provided as source code. |
| Software, algorithm | CHOPCHOP | | https://chopchop.cbu.uib.no/ | |
| Other | PDMS Sylgard 184 | Dow Corning | #000105989377 | For further instructions, see PMID:23681634 |
| Other | 1-Octadecylmer captan | Sigma-Aldrich | #O1858 | 1.5 mM in EtOH |
| Other | HS-C11-EG6-OH | ProChimia | #TH-001-m11.n6 | 1 mM in EtOH |
| Other | TPE-TA TH-001-m11.n6 | Sigma-Aldrich | #409073 | PMID:32967835 |
| Other | PETA | Sigma-Aldrich | # 246794 | PMID:32967835 |
| Other | Host-Guest system-based hydrogel | *Hippler et al., 2020* | | For detailed descriptions and composition, see PMID:32967835 |
| Other | Alexa Fluor 488 or Alexa Fluor 647 coupled phalloidin | ThermoFisher Scientific | #A12379 and #A22287 | |

## Cell culture

U2OS WT cells were obtained from the American Type Culture Collection (Manassas, USA). U2OS NM II-KO cell lines and U2OS GFP-NM IIA or B knock-in cell lines were generated as described in the following sections. All cell lines were tested for mycoplasma infection with negative results. For routine cultivation, cells were passaged every 2–3 days and maintained in DMEM (Pan-Biotech #P04-03590) supplemented with 10% bovine growth serum (HyClone #SH3054.03) at 37°C under a humidified atmosphere containing 5% $CO_2$. Cells, plated on FN-coated coverslips or micropatterned substrates, were allowed to spread for 3 hr, cells in 3D micro-scaffolds for 2 hr.

## Generation of NM II-KO and GFP-NM II knock-in cell lines

CRISPR/Cas9 was used to generate knock-out and knock-in cell lines. Guide sequences for the respective protein of interest were determined using the online tool 'CHOPCHOP' (https://chop-chop.cbu.uib.no/). All used guide sequences are depicted in 5′-to-3′ direction in *Table 2*. Oligos for gRNA construction were obtained from Eurofins genomics (Ebersberg, Germany). NM II-KO cell lines were generated according to the guidelines in *Ran et al., 2013*, using the single plasmid system from Feng Zhang's lab (Addgene #62988). All known splice variants of NMHC IIB and NMHC IIC were targeted by the respective sgRNA. To select for transfected cells, 5 µg mL$^{-1}$ puromycin was added 48 hr post transfection to the culture medium and the cells were selected for another 48 hr. Single cells were derived by limiting dilution and cell colonies were screened for indels and loss of protein expression.

Fluorescent knock-in cell lines with GFP fused to the N-terminus of NMHC IIA or NMHC IIB were generated according to the guidelines in *Koch et al., 2018*. Briefly, a paired Cas9D10A nickase approach (*Ran et al., 2013*) was used to generate a double strand break in close proximity of the first coding exon (exon 2) of *MYH9* or *MYH10*. The guide sequences were cloned into pX335-U6-

**Table 2.** Used gRNA and primer sequences.

| Sequence (5 '→ 3') | Target Gene | Description |
|---|---|---|
| GCACGTGCCCTCAACGAAGCCT | MYH9 | gRNA for DSB in |
| GCTGAAGGATCGCTACTATTC | MYH10 | gRNA for DSB in Exon 2 |
| GCGGAGTAGTACCGCTCCCGG | MYH14 | gRNA for DSB in Exon 2 |
| GCTTATAGCCAGGACCTAAGC | MYH9 | gRNA for SSN in Exon 2 |
| GTGCCGATAAGTATCTCTATG | MYH9 | gRNA for SSN in Exon 2 |
| GCAATTGCCTCTAAGAGAAG | MYH10 | gRNA for SSN in Exon 2 |
| GGCGCAGAGAACTGGACTCG | MYH10 | gRNA for SSN in Exon 2 |
| GCAAAGAGAAGAGGTGTGAGC | MYH9 | Primer (fwd)_NM IIA-KO |
| AGTTCAAGGATGTCACCCCA | MYH9 | Primer (rev)_NM IIA-KO |
| GTTAGTATGGCTGTGAAGAGGT | MYH10 | Primer (fwd)_NM IIB-KO |
| TCAAAGAAAAGCAAGACATGGGT | MYH10 | Primer (rev)_NM IIB-KO |
| AAGAAAGTTGTGCAGCCTGG | MYH9 | Primer binding in LHA |
| GAGCCCTGAGTAGTAACGCT | MYH9 | Primer binding in RHA |
| CATGTTTCTTGGAACCTGGCA | MYH9 | Primer_5'region of LHA_MYH9 |
| GCAAACCCATCAGACAACCA | MYH10 | Primer binding in LHA |
| ATTCTCTGCCAACTCCACCA | MYH10 | Primer binding in RHA |
| CCTCTGCTAGCCCTTTGTGA | MYH10 | Primer_5'region of LHA_MYH10 |
| GATGTTGCCGTCCTCCTTGA | GFP | Primer (rev)_GFP |

Abbreviations: DSB = Double strand break; SSN = Single strand nick; fwd = forward; rev = reverse; LHA = Left homology arm; RHA = Right homology arm.

Chimeric_BB-CBh-hSpCas9n(D10A) (*Cong et al., 2013*). The plasmid was a gift from Feng Zhang (Addgene #42335).

U2OS WT cells were transfected with the according sgRNAs and donor plasmids. GFP-positive cells were sorted using an FACSAria II cell sorter (BD Biosciences). Single cells were derived by limiting dilution and cells were screened for correct insertion of the eGFP by DNA sequence analysis, western blot and immunofluorescence.

## Western blotting

A confluent monolayer of cells in a six-well plate was lysed in 150 µl ice-cold lysis buffer (187 mM Tris/HCl, 6% SDS, 30% sucrose, 5% β-mercaptoethanol), heated at 95°C for 5 min and centrifuged at 13.000 rpm for 10 min. Thirty µl of the supernatant was loaded onto an 8% gel. The proteins were resolved by SDS-PAGE and transferred to a PVDF membrane by tank blotting at 150 mA for 2 hr using the Miniprotean III System from Bio-Rad (Hercules, USA). The membrane was blocked for 1 hr with 5% skim milk in PBS containing 0.05% Tween-20. The following antibody incubation steps were also carried out in the blocking solution. Primary antibodies were applied over night at 4°C and secondary antibodies for 2 hr at room temperature. Between the antibody incubation steps, membranes were washed in PBS/Tween-20. Following primary antibodies were used: mouse monoclonal to α-Tubulin (Sigma-Aldrich #T5168), rabbit polyclonal to NMHC IIA (BioLegend, #909801), rabbit polyclonal to NMHC IIB (BioLegend, #909901), rabbit monoclonal to NMHC IIC (CST, #8189S), rabbit polyclonal to GFP (Abcam, #ab6556). Secondary horseradish peroxidase-coupled anti-mouse or anti-rabbit antibodies were from Jackson Immunoresearch (#711-036-152 and #715-035-150). The membranes were developed with the SuperSignal West Pico PLUS chemiluminescent substrate (Thermo-Fisher Scientific #34579) according to manufacturer's instructions. Signal detection was carried out using an Amersham Imager 600 from GE Healthcare (Chicago, USA).

## Sequence analysis

gDNA was isolated using the DNeasy Blood and Tissue Kit (Qiagen #60506) and the target region was amplified via PCR. Primers were designed using the Primer3 freeware tool (*Untergasser et al., 2012*) and purchased from Eurofins genomics (Ebersberg, Germany). All used primers are listed in *Table 2*. PCR products were either cloned into the pCR II-Blunt-TOPO vector using the Zero Blunt TOPO PCR cloning kit (ThermoFisher Scientific #K2875J10) for subsequent sequencing or sequenced directly. Sequencing was carried out at LGC Genomics (Berlin, Germany) and the results were compared to WT sequences using the free available version of SnapGene Viewer (https://www.snap-gene.com/snapgene-viewer/).

## Transfection and constructs

Transfections were carried out using Lipofectamine 2000 (ThermoFisher Scientific #11668027) according to manufacturer's instructions. The cells were transfected 48 hr prior to the experiment. CMV-GFP-NMHC IIA (Addgene #11347) and CMV-GFP-NMHC IIB (Addgene #11348) were gifts from Robert Adelstein (*Wei and Adelstein, 2000*). NMHC IIB-mApple was a gift from Michael Davidson (Addgene #54931). pSPCas9(BB)−2A-Puro (PX459) V2.0 (Addgene #62988) and pX335-U6-Chimeric_BB-CBh-hSpCas9n(D10A) (Addgene #42335) were gifts from Feng Zhang. Guide sequences for the generation of NMHC II depleted cells or GFP knock-in cells were introduced by digesting the plasmids with BbsI and subsequent ligation (*Ran et al., 2013*). pMK-RQ-*MYH9* and pMK-RQ-*MYH10* donor plasmids for homology directed repair were constructed by flanking the coding sequence of eGFP with 800 bp homology arms upstream and downstream of the double strand break near the start codon of *MYH9* or *MYH10* exon two and the plasmids were synthesized by GeneArt (ThermoFisher Scientific).

## Fabrication of micropatterned substrates

Micropatterned substrates were prepared using the microcontact printing technique (*Mrksich and Whitesides, 1996*). Briefly, a master structure, which serves as a negative mold for the silicon stamp was produced by direct laser writing (*Anscombe, 2010*) and the stamp was molded from the negative using Sylgard 184 (Dow Corning #105989377). The stamp-pattern resembles a sequence of crosses with different intersections, a bar width of 5 µm and edge length of 45–65 µm. The pattern was either transferred using gold-thiol chemistry (*Mrksich et al., 1997*) or direct microcontact printing (*Fritz and Bastmeyer, 2013*). When using gold-thiol chemistry, the stamp was inked with a 1.5 mM solution of octadecylmercaptan (Sigma Aldrich #O1858) in ethanol and pressed onto the gold-coated coverslip, forming a self-assembled monolayer at the protruding parts of the stamp. For the subsequent passivation of uncoated areas, 2.5 mM solution of hexa(ethylene glycol)-terminated alkanethiol (ProChimia Surfaces #TH-001-m11.n6) in ethanol was used. Micropatterned coverslips were functionalized with a solution of 10 µg ml$^{-1}$ FN from human plasma (Sigma Aldrich #F1056) for 1 hr at room temperature. For direct microcontact printing, stamps were incubated for 10 min with a solution of 10 µg ml$^{-1}$ FN and pressed onto uncoated a coverslip. Passivation was carried out using a BSA-Solution of 10 mg ml$^{-1}$ in PBS for backfilling of the coverslip at room temperature for 1 hr.

## Fabrication of stimuli-responsive 3D micro-scaffolds

The fabrication and characterization of the stimuli-responsive 3D micro-scaffolds was described in detail in *Hippler et al., 2020*. Briefly: A commercial direct laser writing system (Photonic Professional GT, Nanoscribe GmbH) equipped with a 63×, NA = 1.4 oil-immersion objective was used for the fabrication process. In three consecutive writing steps, the various components of the micro-scaffolds were produced by polymerizing liquid photoresists in the voxel of a femto-second pulsed near infrared laser. By using different photoresists that possess hydrophilic or hydrophobic surface properties after polymerization, protein-repellent or protein-adhesive substructures were created. TPETA photoresist was used to write the protein-repellent walls and PETA photoresist for the protein-adhesive beams. For the stimuli-responsive hydrogel, a host-guest based photoresist was polymerized in the center of the scaffold. All mixtures and reagents can be found in *Hippler et al., 2020*. Before the sample was used for further experiments, it was immersed overnight in water with 20 mM 1-Adamantanecarboxylic acid (Sigma-Aldrich #106399). This solution triggered the swelling of the hydrogel that helped to remove unpolymerized residues from the material network.

## Immunostaining

Samples were fixed for 10 min using 4% paraformaldehyde in PBS and cells were permeabilized by washing three times for 5 min with PBS containing 0.1% Triton X-100. Following primary antibodies were used: mouse monoclonal to FN (BD Biosciences, #610078), rabbit polyclonal to NMHC IIA (BioLegend, #909801), rabbit polyclonal to NMHC IIB (BioLegend, #909901), rabbit monoclonal to NMHC IIC (CST, #8189S), mouse monoclonal to Paxillin (BD Biosciences, #610619), mouse monoclonal to pRLC at Ser19 (CST, #3675S). All staining incubation steps were carried out in 1% BSA in PBS. Samples were again washed and incubated with fluorescently coupled secondary antibodies and affinity probes. Secondary Alexa Fluor 488-, Alexa Fluor 647- and Cy3-labeled anti-mouse or anti-rabbit antibodies were from Jackson Immunoresearch (West Grove, USA). F-Actin was labeled using Alexa Fluor 488- or Alexa Fluor 647-coupled phalloidin (ThermoFisher Scientific #A12379 and #A22287) and the nucleus was stained with DAPI (Carl Roth #6335.1). Samples were mounted in Mowiol containing 1% N-propyl gallate.

## Fluorescence imaging

Images of immunolabeled samples on cross-patterned substrates were taken on an AxioimagerZ1 microscope (Carl Zeiss, Germany). To obtain high-resolution images of minifilaments, the AiryScan Modus of a confocal laser scanning microscope (LSM 800 AiryScan, Carl Zeiss) or a non-serial SR-SIM (Elyra PS.1, Carl Zeiss) were used. The grid for SR-SIM was rotated three times and shifted five times leading to 15 frames raw data of which a final SR-SIM image was calculated with the structured illumination package of ZEN software (Carl Zeiss, Germany). Channels were aligned by using a correction file that was generated by measuring channel misalignment of fluorescent TetraSpeck microspheres (ThermoFischer, #T7280). All images were taken using a 63×, NA = 1.4 oil-immersion objective.

For live cell flow measurements, the incubation chamber was heated to 37°C. Cells were seeded on FN-coated cell culture dishes (MatTek #P35G-1.5–14 C) or micropatterned substrates 3 hr prior to imaging. During imaging, the cells were maintained in phenol red-free DMEM with HEPES and high glucose (ThermoFisher Scientific #21063029), supplemented with 10% bovine growth serum and 1% Pen/Strep.

## Flow analysis and intensity measurements

Flow in vSF or peripheral actin arcs was measured by creating kymographs from a ROI using the reslice function in ImageJ. From these kymographs, movement of individual, persistent minifilaments was tracked manually to determine the flow rate in nm/min.

Quantification of pRLC-, NMHC II-, and GFP-intensities were carried out by calculating the mean intensity along segmented SFs.

## AFM nanoindentation experiments

We used a NanoWizard AFM (JPK Instruments) equipped with a soft silicon nitride cantilever (MLCT, Bruker) with a nominal spring constant of 0.03 N/m to perform the indentation experiments. The cells were allowed to spread for at least 8 hr in cell culture dishes (TPP # 93040) before the experiment was performed. For each cell, 16 individual measurements were performed above the nuclear region. A Hertz model was fitted to the resulting force displacement curves and the resulting Young's moduli were averaged.

## Quantification of FA parameters and $R(d)$-correlations

Quantification of FAs was performed using the pixel classification functionality of the image analysis suite ilastik (*Berg et al., 2019*). First, ilastik was trained to mark the cell area. In a separate classification project ilastik was trained to discern between FA and non-FA. The segmentations were exported in the npy file format for analysis in custom scripts. To determine the number of FAs connected component analysis was applied to the segmented FAs as implemented in openCV 3.4.1.

Quantifications of $R(d)$-correlations were carried out by manually fitting circles to the peripheral actin arcs of cells on cross-patterned substrates. The spanning distance $d$ was defined as the cell area covering the passivated substrate area. In cases, where the cell was polymerizing actin along the functionalized substrate without surpassing the complete distance to the cell edges (as observed

in the case of NM IIA-KO cells), only the distance of the cell body covering the passive substrate was considered.

## Stretching experiments and force measurements

Live cell imaging was performed as described above using an LSM 800 equipped with a 40×, NA = 1.2 water-immersion objective and the motorized mechanical stage to sequentially move to all the positions during the time series. To exchange solutions during the experiment, the sample was mounted in a self-built fluidic chamber. For initial force measurements, cells were imaged for 30–60 min under steady-state conditions and then detached from the substrate using Trypsin/EDTA. For stretching experiments, the cells were first imaged for 10 min under steady-state conditions. To induce the mechanical stretch, the solution was exchanged to medium containing 20 mM 1-Adamantanecarboxylic acid and the cells were imaged in the stretched state for 30–70 min. Releasing the stretch was obtained by again replacing the medium with normal imaging medium and the cellular reaction was monitored for up to 30–50 min.

To calculate the initial and reactive forces, the images were analyzed by digital image cross-correlation based on a freely available MATLAB code (*Eberl et al., 2006*) (MathWorks) that was customized as described in *Hippler et al., 2020*. The code is available as source code file. In every scene, regions of interest were defined on the four beams and every frame of the time series was compared to a reference image at t = 0. The calculation of the maximum cross-correlation function resulted in the 2D local displacement vector. Four different positions per beam were tracked and averaged to obtain a mean displacement per beam as a function of time. Additional tracking of solid marker structures and reference scaffolds without cells was used to correct potential offsets and deviations that are not induced by cellular forces. Ultimately, the measured displacements were converted to cell forces by modeling the properties of the micro-scaffolds by finite element calculations (see [*Hippler et al., 2020*] for details).

## Cell-seeded collagen gels

CSCGs were generated according to the guidelines in *Provenzano et al., 2010*. We used collagen I from rat tails (Enzo Life Sciences #ALX-522–435) and seeded $1.5 \times 10^5$ cells in collagen matrices with a final concentration of 1 mg/ml. As a neutralizing buffer, 0.1 M HEPES in 2×PBS was used in equal volumes to the collagen solution. 250 µl total volume were distributed in 18 mm glass bottom dishes and allowed to polymerize at 37°C. After 2 hr, the dish was backfilled with DMEM and the CSCG's were cultivated in suspension for another 18 hr. After 20 hr total incubation time, CSCG's were fixed in 4% PFA and the diameter was measured.

For gels fixed to the glass bottom, dishes were pre-coated with thin plate coating collagen I (Enzo Life Sciences #ALX-522-440-0050) and allowed to dry overnight. On the following day, CSCG's were fabricated as described and polymerized on the pre-coated culture dishes.

## Modeling

For more information about the dTEM and the parameters, we refer the reader to Appendix 1, where a detailed description can be found.

## Acknowledgements

We thank Alisha Rapp (KIT) for her help with the analysis of GFP- and pRLC intensities. This work is supported by the Deutsche Forschungsgemeinschaft (DFG, German Research Foundation) under Germany's Excellence Strategy through EXC 2082/1-390761711 (the Karlsruhe-Heidelberg 3DMM2O Excellence Cluster, to USS and MB) and EXC 2181/1–390900948 (the Heidelberg STRUCTURES Excellence Cluster, to USS). USS is a member of the Interdisciplinary Center for Scientific Computing (IWR) at Heidelberg. JG acknowledges support by the Research Training Group of the Landesstiftung Baden-Württemberg on Mathematical Modeling for the Quantitative Biosciences.

# Additional information

## Funding

| Funder | Grant reference number | Author |
| --- | --- | --- |
| Deutsche Forschungsgemeinschaft | EXC 2082/1-390761711 | Ulrich Sebastian Schwarz<br>Martin Bastmeyer |
| Deutsche Forschungsgemeinschaft | EXC 2181/1 - 390900948 | Ulrich Sebastian Schwarz |

The funders had no role in study design, data collection and interpretation, or the decision to submit the work for publication.

## Author contributions

Kai Weißenbruch, Conceptualization, Data curation, Formal analysis, Validation, Visualization, Methodology, Writing - original draft, Writing - review and editing; Justin Grewe, Conceptualization, Data curation, Software, Formal analysis, Validation, Methodology, Writing - original draft, Writing - review and editing; Marc Hippler, Conceptualization, Data curation, Software, Validation, Investigation, Visualization, Methodology; Magdalena Fladung, Data curation, Validation, Visualization; Moritz Tremmel, Kathrin Stricker, Data curation, Validation; Ulrich Sebastian Schwarz, Conceptualization, Supervision, Funding acquisition, Investigation, Writing - original draft, Project administration, Writing - review and editing; Martin Bastmeyer, Conceptualization, Resources, Supervision, Funding acquisition, Investigation, Writing - original draft, Project administration, Writing - review and editing

## Author ORCIDs

Kai Weißenbruch ![ID] https://orcid.org/0000-0002-9463-6725
Magdalena Fladung ![ID] http://orcid.org/0000-0002-4213-2891
Moritz Tremmel ![ID] http://orcid.org/0000-0001-8901-9362
Ulrich Sebastian Schwarz ![ID] https://orcid.org/0000-0003-1483-640X
Martin Bastmeyer ![ID] http://orcid.org/0000-0003-3471-8400

## Decision letter and Author response

Decision letter https://doi.org/10.7554/eLife.71888.sa1
Author response https://doi.org/10.7554/eLife.71888.sa2

# Additional files

## Supplementary files

- Source data 1. Raw data sets for quantitative evaluation.

- Source data 2. Western Blot raw images.

- Transparent reporting form

## Data availability

All data generated or analysed during this study are included in the manuscript and supporting files. Source files with the raw data are provided for all Figures, where quantifications are carried out.

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

## Appendix 1

### Crossbridge cycle model for nonmuscle myosin II

In order to model the crossbridge cycle of the different isoforms of nonmuscle myosin II, we consider their three main mechanochemical states and the stochastic transitions between them, as depicted in *Appendix 1—figure 1A*. Our three-state model has been extensively tested and parametrized before (*Erdmann et al., 2013*; *Erdmann et al., 2016*; *Erdmann and Schwarz, 2012*; *Grewe and Schwarz, 2020a*) and is used here with a small modification. In brief, myosin heads bind from the unbound state to the actin filament with rate $k_{01} = 0.2\,\mathrm{s}^{-1}$. Recently it has been shown that this binding occurs in two steps, with a non-stereospecific state existing before the weakly bound state, which is stereospecific. The non-stereospecific intermediate state becomes relevant in the case of myosin II inhibition by blebbistatin (*Rahman et al., 2018*) which we do not consider here, but this observation motivates us to choose a larger value $k_{10} = 0.4\,\mathrm{s}^{-1}$ for the unbinding rate from the weakly bound to the unbound state (rather than $0.004\,\mathrm{s}^{-1}$ as used before [*Erdmann et al., 2013*; *Erdmann et al., 2016*; *Erdmann and Schwarz, 2012*; *Grewe and Schwarz, 2020b*]). From the weakly bound state, the powerstroke occurs with the high rate $k_{12} = 1.4 \cdot 10^6\,\mathrm{s}^{-1}$. Furthermore, the powerstroke is associated with swinging of the lever arm, which is simulated by increasing the individual motor strain $x_i$ by the powerstroke distance $d = 8$ nm. After having performed the powerstroke, myosin can either return to the weakly bound state with the relatively small rate $k_{21} = 0.7\,\mathrm{s}^{-1}$, or it can unbind from actin with a force- and isoform-dependent rate

$$k_{20}^{a/b}(F) = k_{20}^{0a/0b}[\Delta_c \exp(-k_m x_i/f_c) + (1-\Delta_c)\exp(k_m x_i/f_s)]. \tag{A1}$$

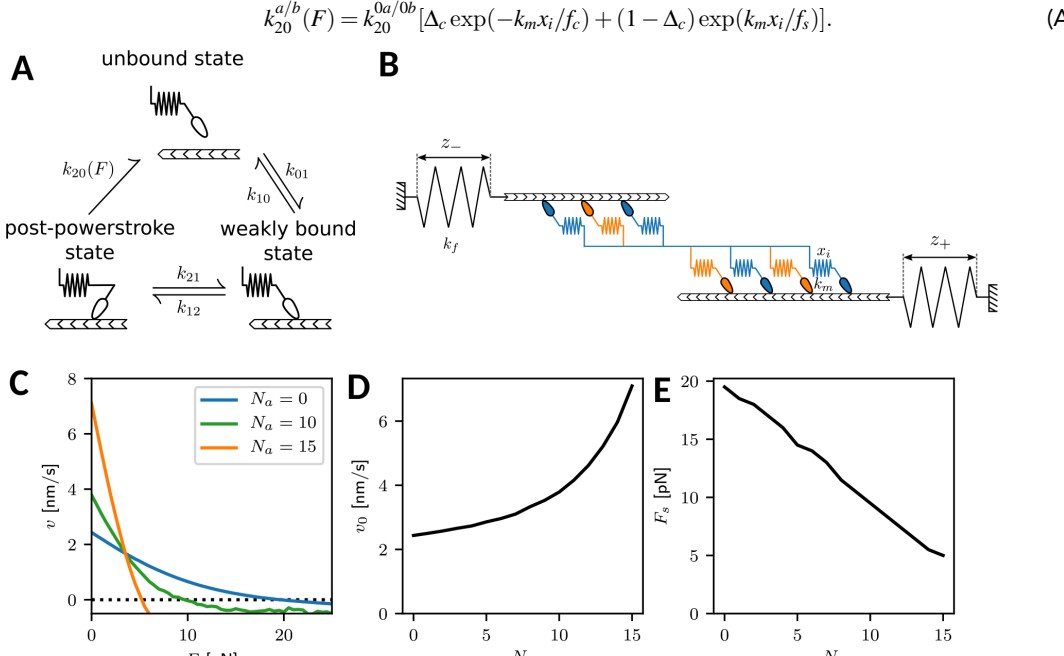

**Appendix 1—figure 1.** Crossbridge cycle model overview. (**A**) Mechanochemical crossbridge cycle for myosin II. (**B**) Tug-of-war of two mixed motor ensembles working against external springs. (**C**) Force-velocity relation for an ensemble (one half of a minifilament) with $N = 15$ motors with varying numbers $N_a$ of NM IIA motors. The number of NM IIB motors is $N_b = N - N_a$. (**D**) Force-free velocity $v_0$ and (**E**) stall force $F_s$ as a function of $N_a$.

Here, $k_{20}^{0a/0b}$ are the transition rates at zero force of the A- and B-isoform, respectively. In particular, the rate for isoform A $k_{20}^{0a} = 1.71\,\mathrm{s}^{-1}$ is much larger than the rate for isoform B $k_{20}^{0b} = 0.35\,\mathrm{s}^{-1}$, which constitutes the mechanochemical difference between isoforms A and B in our model. The two terms in the brackets represent two different unbinding pathways, namely the *catch-path* and the *slip-path*, respectively. Transitions along the catch-path become slower with increasing force with force scale $f_c = 1.66$ pN. The force on the motor is calculated by the motor stiffness $k_m$ and the motor strain $x_i$. Conversely, transitions along the slip-path become faster with increasing force with force scale $f_s = 10.35$ pN. $\Delta_c$ is the fraction of transitions following the catch-path at zero force, while

the complement is the fraction of transitions following the slip-path. Together, these two pathways model a *catch-slip bond*, so dissociation decreases with force at low loading (*catch*) and increases again at high loading (*slip*). Force dependence of the rates is only taken into account if the motor is loaded against its direction of movement, the transition rate defaulting to $k_{20}^0$ for forces in the other direction. Note that in principle the force scales could also be modeled to depend on the isoform. In-vitro experiments suggest that the catch-path force scale for isoform B are lower than for isoform A (*Kovács et al., 2007*). Here, they are kept equal for simplicity, as the difference in the rate at zero force is the dominating effect at low forces.

In a minifilament, the different motor heads are mechanically coupled to form a bipolar structures with two ensembles pulling in opposite directions. This tug-of-war situation is depicted in *Appendix 1—figure 1*. We assume that in each minifilament half, $N = 15$ motor heads are active (*Grewe and Schwarz, 2020a*). They work against external springs with strains $z_-$ and $z_+$, respectively. In addition, each side of the minifilament consists of a variable number of NM IIA and NM IIB motors, $N_a^-$, $N_b^-$, $N_a^+$ and $N_b^+$, respectively, with $N_a^- + N_b^- = N$ and $N_a^+ + N_b^+ = N$. At all times the forces acting on the myosin heads of each sides are balanced against the forces in the external springs. This also yields the value for the motor strains $x_i$ required to evaluate *Equation A1*.

All model parameters are summarized in *Appendix 1—table 1*. The stochastic model is simulated using the Gillespie algorithm, which uses the assumed rates to determine the time until the next reaction takes place (*Gillespie, 1976*). From these simulations we obtain the force-velocity relations of the minifilaments with variable isoform content by averaging over many individual trajectories. As shown in *Appendix 1—figure 1C*, we obtain well-defined relations in all cases, which allow us to predict the free velocity $v_0$ and the ensemble stall force $F_s$. We find that with increasing A-content, the free velocity $v_0$ increases and the stall force $F_s$ decreases, as shown in *Appendix 1—figure 1D and E*, respectively. This demonstrates that minifilament with more A-isoforms are more dynamic, but also less stable mechanically. As a third important quantity, we can extract an effective friction coefficient $\xi_m$ from $v'(F_s) = -1/\xi_m$, which is the steepness of the force-velocity relation at the stall force. This friction coefficient $\xi_m$ decreases with increasing A-content, reflecting that the system becomes more dynamic.

**Appendix 1—table 1.** Model parameters.

| Parameter | Symbol | Value | References |
|---|---|---|---|
| Transition rates [s⁻¹] | $k_{20}^{a0}$ | 1.71 | *Erdmann et al., 2016*; *Stam et al., 2015* |
| | $k_{20}^{b0}$ | 0.35 | *Erdmann et al., 2016*; *Stam et al., 2015* |
| | $k_{01}$ | 0.2 | *Erdmann et al., 2016*; *Stam et al., 2015* |
| | $k_{10}$ | 0.4 | *Rahman et al., 2018* |
| | $k_{12}$ | $4.10^6$ | *Vilfan and Duke, 2003*; *Erdmann et al., 2016* |
| | $k_{21}$ | 0.7 | *Vilfan and Duke, 2003*; *Erdmann et al., 2016* |
| Catch-path fraction | $\Delta_c$ | 0.92 | *Erdmann et al., 2016*; *Stam et al., 2015* |
| Catch-path force scale [pN] | $f_c$ | 1.66 | *Erdmann et al., 2016*; *Stam et al., 2015* |
| Slip-path force scale [pN] | $f_s$ | 10.35 | *Erdmann et al., 2016*; *Stam et al., 2015* |
| Neck-linker stiffness [pN/nm] | $k_m$ | 0.7 | *Erdmann et al., 2016*; *Stam et al., 2015* |
| Powerstroke distance [nm] | d | 8 | *Vilfan and Duke, 2003*; *Erdmann et al., 2013*; *Erdmann and Schwarz, 2012* |
| External springs [pN/nm] | $k_f$ | 4 | *Albert et al., 2014* |

## Dynamic tension elasticity model (dTEM)

The invaginated shapes of strongly adherent cells have been shown before to result from actomyosin contractility (*Zand and Albrecht-Buehler, 1989*; *Bischofs et al., 2008*; *Bar-Ziv et al., 1999*). From a geometrical viewpoint, there are two actomyosin-related forces which balance each other along the invaginated arcs: an isotropic surface tension $\sigma$ in the cortex and a line tension $\lambda$ in the peripheral fiber. This leads to the Laplace law $R = \lambda/\sigma$ for the arc radius (*Bischofs et al., 2008*; *Bar-Ziv et al., 1999*). Because experimentally it has been found that the radius $R$ also depends on the spanning distance $d$ of the invaginated arc, the line tension $\lambda$ has been argued to also contain an elastic contribution (tension-elasticity model, TEM), leading to $\lambda(d)$ and an increasing $R$-$d$ relation, as observed experimentally (*Bischofs et al., 2008*). The line tension $\lambda$ can be identified with the force $F$ that acts in the fiber and in particular on the focal adhesions (FAs) by which it is anchored. Here we have confirmed the increasing $R$-$d$ relation and in addition revealed that it also depends on isoform content. In order to make contact to the different crossbridge cycles of isoform A versus B, we now introduce a dynamic variant of the TEM (dTEM).

We consider a peripheral fiber of length $L$ that is contracted by minifilaments that are distributed uniformly along its length. The contraction speed due to contractility is denoted by $v_c$ and should depend on both the force $F$ inside the fiber and its length $L$ (like for muscle, a longer fiber should contract faster). In order to be able to obtain a stationary situation, new length has to be generated in the FAs at which the fiber is anchored, as observed experimentally as flow out of the FAs (*Endlich et al., 2007*). The corresponding polymerization velocity is named $v_p$ and also should depend on fiber force $F$, but not on $L$, because it is a local property of the FAs. Specifically, it has been shown that the main polymerization factor in FAs, the formin mDia1, increases its polymerization rate with force (*Jégou et al., 2013*). Together, the two velocities lead to a length change

$$\dot{L} = v_p(F) - v_c(F,L) \tag{A2}$$

of the stress fiber, which is the central dynamic equation that we will analyze here.

The contractile speed of molecular motors can be modeled by a linearized force-velocity relation

$$v_m(F) = \frac{1}{\xi_m}(F_s - F) \tag{A3}$$

with motor stall force $F_s$ and effective friction coefficient $\xi_m$ for a sarcomeric unit of reference length $L_0$ (for a linear force-velocity relation, two parameters are sufficient and the free velocity follows as $v_0 = F_s/\xi_m$). Linear scaling of contraction speed with stress fiber length implies for a stress fiber of length $L$

$$\frac{v_m}{L_0} = \frac{v_c(F,L)}{L} \Rightarrow v_c(F,L) = \frac{L}{L_0\xi_m}(F_s - F). \tag{A4}$$

This relates the contraction speed in the fiber to the properties of the motor ensemble. For the polymerization speed of actin at the focal adhesions we assume a linear force dependence

$$v_p = \frac{F}{\xi_f} \tag{A5}$$

where $\xi_f$ is the effective friction coefficient inside the FAs.

The line tension that we obtain from the interplay of contraction and polymerization can now be related to the circular arc radius $R$ with the Laplace law

$$R = \frac{F}{\sigma}. \tag{A6}$$

This means that the dependence on $F$ in *Equation (A2)* is in fact a dependence on the radius of curvature $R$, that in turn will depend on the length of the stress fiber $L$ and the spanning distance $d$. To close *Equation (A2)*, a geometrical equation that relates these quantities is required.

Stress fiber length $L$ and the radius $R$ are trivially related by the central angle $\varphi$ as $L = R\varphi$. The central angle is in turn dependent on spanning distance $d$ and radius of curvature $R$. We then obtain

$$L = \begin{cases} 2R\arcsin\left(\frac{d}{2R}\right) & 0 \leq \varphi \leq \pi \\ 2R\left(\pi - \arcsin\left(\frac{d}{2R}\right)\right) & \pi \leq \varphi \leq 2\pi \end{cases} \tag{A7}$$

which is an implicit definition for $R(L,d)$. While circular arcs with larger central angle than $\pi$ cannot be observed on cross patterns, we do observe them on homogeneous substrates and in collagen gels. Therefore they are also included here in our discussion. In *Appendix 1—figure 2A* we plot $R$ as a function of $L$ as defined by *Equation (A7)* after rescaling all lengths with $d$. We see that the inversion is not unique and has two branches. This is due to $d(L)$ not being uniquely defined by *Equation (A7)*, which is visualized in *Appendix 1—figure 2B*. For large $L/d$, the angle $\varphi$ is small, $R/d$ is large and $L \approx d$. For $\varphi = \pi$ (half-circle), we have $L/d = \pi/2$ and $R = d/2$. For larger $\varphi$, $R/d$ increases again and finally diverges. On the cross patterns, we typically deal with the small angle case.

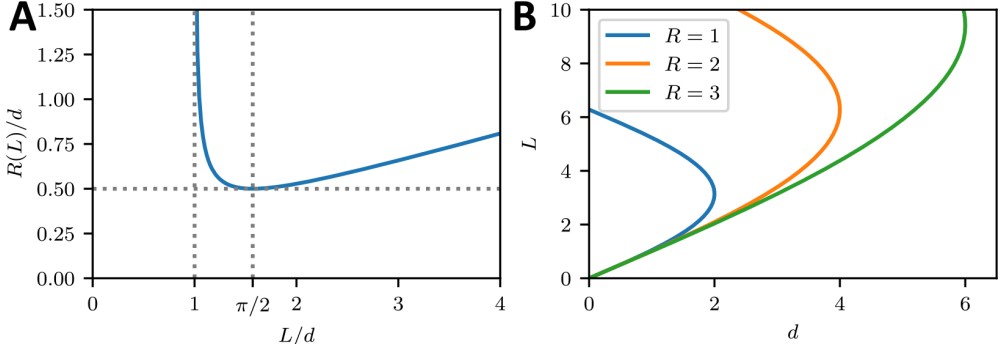

**Appendix 1—figure 2.** Geometrical relations between R, L and d. (**A**) The arc radius $R(L)$ is a function of arc length $L$ on the interval $L \in (d, \infty)$. For $L/d < \pi/2$ it is monotonically decreasing and for $L/d > \pi/2$ it is monotonically increasing. Normalizing $R$ and $L$ to the spanning distance $d$ yields a universal result. (**B**) Due to the geometry, $L(d)$ is not a well defined function, as for each spanning distance there exist two solutions for a central angle $0 < \varphi < 2\pi$.

By combining *Equation (A2), (A4), (A5), (A6)*, we now obtain our central dynamical equation:

$$\dot{L} = -\frac{L}{L_0 \xi_m}(F_s - \sigma R(L,d)) + \frac{(L,d)}{\xi_f}. \tag{A8}$$

We non-dimensionalize this equation by measuring distance in units of $R_{\max} = F_s/\sigma$ and time in units of $\tau = \xi_f/\sigma$:

$$\dot{l} = -\frac{l}{l_m}[1 - r(l,\delta)] + r(l,\delta) \tag{A9}$$

where $r$ is the dimensionless arc radius and $\delta$ is the dimensionless spanning distance. Moreover we have defined

$$l_m = \frac{\xi_m L_0}{\xi_f R_{max}} = \frac{\xi_m L_0 \sigma}{\xi_f F_s}. \tag{A10}$$

*Appendix 1—figure 3* shows $\dot{l}$ as a function of $l$. We see that for sufficiently small values of $\delta$ and $l_m$, both a stable and an unstable fixed point exist, at small and large values of $l$, respectively. The stable fixed point at small $l$ corresponds to a steady state. This state is lost through a saddle-node bifurcation for larger values of the dimensionless quantities $\delta$ and $l_m$.

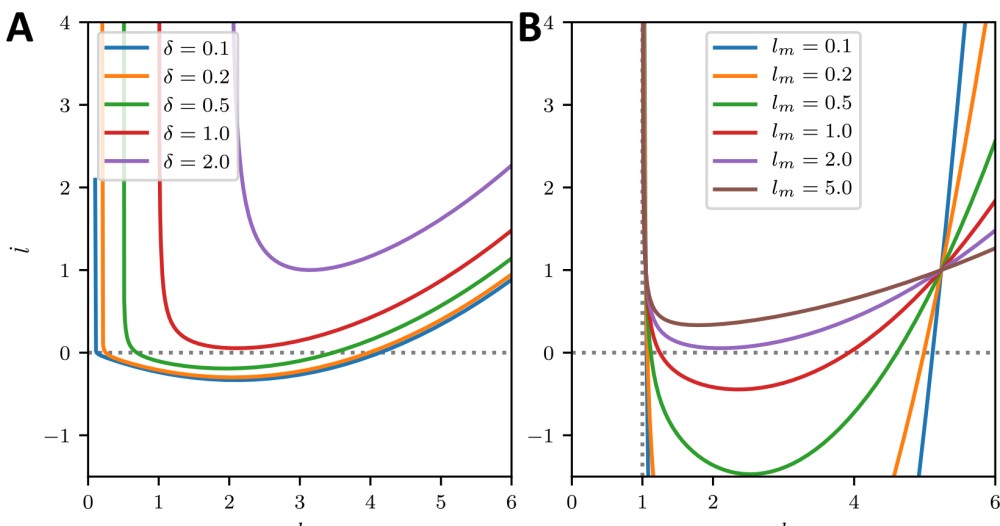

**Appendix 1—figure 3.** Stability analysis. (**A**) Change in dimensionless arc length $\dot{l}$ for a range of spanning distances $\delta$ as a function of $l$ for $l_m = 2$. For small spanning distances there are two steady states. The one at lower $l$ is stable, while the other one is unstable. For increasing $\delta$ the system approaches a saddle-node distribution, beyond which both steady states vanish and the length of the peripheral arc increases indefinitely. (**B**) $\dot{l}$ For a range of $l_m$ and spanning distance $\delta = 1$. Again the two fixed points vanish in a saddle node bifurcation.

By solving for the stationary state with $\dot{l} = 0$, we arrive at a relation between the arc radius $R$ and the steady state fiber length $L$, which we give both without and with dimensions:

$$r = \frac{l}{l_m + l}, \quad R = \frac{L}{L_m + L} R_{\max} . \tag{A11}$$

Thus, the arc radius $R$ initially increases linearly with arc length $L$ and then starts to saturate towards $R_{\max} = F_s/\sigma$ at the arc length of half maximal radius $L_m = \xi_m L_0/\xi_f$. Here, the two competing length scales are the Laplace radius $R_{\max}$ and the ratio of the slopes of the length-normalized force-velocity relation of the fiber and the force-velocity relation of the FA.

To compare with our experimental results, we finally have to convert the $R(L)$ relation to a $R(d)$ relation. Practically this can be done best by first fixing $d$, then finding the lower fixed point for $l$ of *Equation (A9)* (if it exists) and finally calculating $R$ from *Equation (A11)*. Alternatively, one can use the approximation $L \approx d$ for small central angles, which we did in the main text. We then arrive at

$$r = \frac{\delta}{\delta_m + d}, \quad R = \frac{d}{d_m + d} R_{\max}, \tag{A12}$$

where we have renamed $l_m = \delta_m$ and $L_m = d_m$. This equation is the central result (1) in the main text. The result for $\delta_m < \pi$ is shown in *Appendix 1—figure 4A*. For the exact case we find $R(d)$ relations that always start at $R(0) = 0$. From there, the function increases monotonically while curving downward and approaching a plateau for low $\delta_m$. In this regime, the approximation is reasonably accurate. At $d_{high}$, where $\varphi = \pi$, the function stops approaching the plateau. Shortly after, at $d_{crit}$, the saddle-node bifurcation occurs and at higher spanning distances $d$ no steady state exists anymore. For the parameters chosen in *Appendix 1—figure 4A*, $d_{high} \approx d_{crit}$ and for this reason, the region where central angles $\varphi > \pi$ is very small.

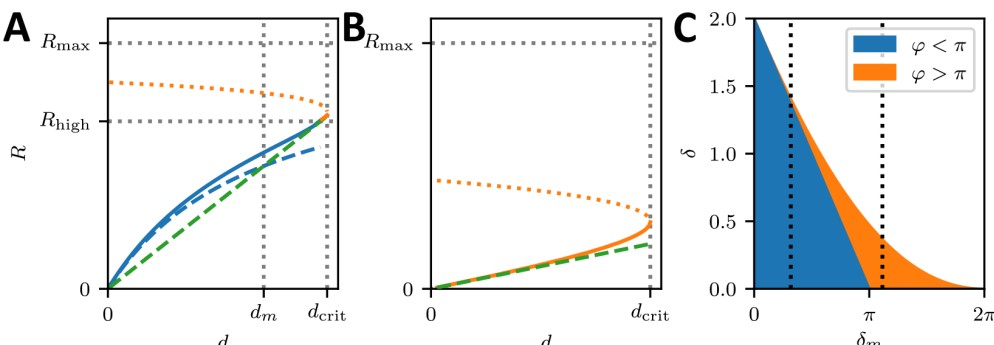

**Appendix 1—figure 4.** Predicted R(d)-correlations. (**A,B**) $R(d)$ relationship predicted by the model. Blue solid lines are related to stable steady state lengths, where the arc is smaller than a half cicle, while dashed blue lines represent the result with approximation $\arcsin x = x$. Solid orange lines represent stable steady states where the central angle $\varphi > \pi$ and dotted orange lines represent unstable steady states. The dashed green line denotes $d = 2R$, which corresponds to the circle of smallest radius by geometry. The colors in the phase diagrams represent parameter regions, where stable solutions can be found and represent arcs that are smaller (blue) and larger (orange) than half circles, respectively. In the white area, the arc curls inward and increases its radius indefinitely. (**A**) $\delta_m = 1$ (**B**) $\delta_m = 3.5$ (**C**) Phase diagram indicating whether the central angle is larger or smaller than $\pi$. The dashed vertical lines visualize the parameter range described by subfigure **A** and **B**, respectively.

For $\delta_m \geq \pi$, the central angle $\varphi > \pi$. Again this cannot be observed on cross patterns, but we did experimentally observe this case in collagen gels. The $R(d)$ relation again starts at zero for zero spanning distance and curves upward until reaching the critical value for $d$ where the steady states do not exist anymore as shown in *Appendix 1—figure 4B*. *Appendix 1—figure 4C* gives a general overview of the parameter range indicating regions of upward curvature connected to central angles lower than $\pi$ together with regions of downward curvature which are connected to central angles higher than $\pi$.

Using $\varphi r = l$ and $\sin \varphi/2 = \delta/2r$ in *Equation (A11)* we find

$$\delta = 2 \sin\frac{\varphi}{2}\left(1 - \frac{\delta_m}{\varphi}\right),$$  (A13)

that is the contour lines of the central angle $\varphi$ as a function of $(\delta, \delta_m)$ are linear functions. In particular for $\varphi = \pi$, which is the situation of highest spanning distance given the radius, we find:

$$d_{\text{high}} = 2\left(R_{\max} - \frac{d_m}{\pi}\right).$$  (A14)

For angles $\varphi > \pi$, the linear functions from *Equation (A13)* however intersect, which contour lines should never do. This implies, that the contour lines for $\varphi$ have to be constrained more carefully. As the steady state central angle $\varphi(\delta_m, \delta)$ is a smooth function that is well defined for low enough $\delta_m$ and $\delta$ and monotonically increases with $\delta_m$ and $\delta$ we can search for the maximum stable spanning distance $\delta$ w.r.t. central angle $\varphi$ using *Equation (A13)*. This yields,

$$\delta_m^{\max} = \frac{\varphi^2 \cos\frac{\varphi}{2}}{\varphi \cos\frac{\varphi}{2} - 2\sin\frac{\varphi}{2}},$$
$$\delta^{max} = \frac{\left(2 \sin\frac{\varphi}{2}\right)^2}{2 \sin\frac{\varphi}{2} - \varphi \cos\frac{\varphi}{2}}$$  (A15)

which marks the border of stability of the system for $\pi \leq \varphi \leq 2\pi$ as shown in *Appendix 1—figure 5*. Contour lines of $\varphi$ for $\varphi > \pi$ start on this curve, with smaller $\delta_m$ having to be excluded from *Equation (A13)*.

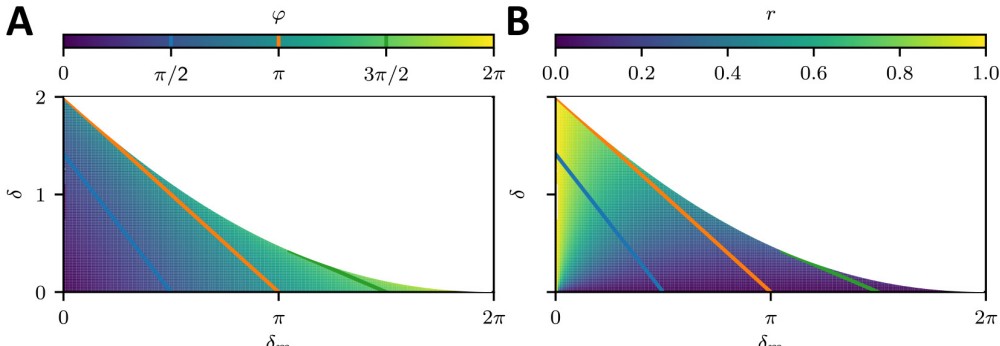

**Appendix 1—figure 5.** Phase diagram. (**a**) Central angle φ and (**b**) Dimensionless radius $r$ as a function of $\delta$ and $\delta_m$. The white region represents unstable parameter configurations, while the colored areas represent the stable steady state central angle and radius respectively. The colored lines are contour lines of the central angle and indicate upper bounds for stability given that the maximum permissible central angle by the micropattern geometry.

Appendix 1—figure 5 shows the overall dependencies of the central angle φ and the radius $r$ on the spanning distance and the respective length scale $\delta_m$ together with contour lines for the central angle $\varphi$, which can also be interpreted as upper bounds for stability given a specific micropattern geometry. The cross shaped micropattern yields maximum central angles of $\pi/2$ (blue line in *Appendix 1—figure 5* and upper bound of the red region in *Figure 3E* of the main text) as illustrated in *Figure 3B* of the main text.

