## [Decision Letter]

**Acceptance summary:**

This study is of interest to researchers studying the actin cytoskeleton, cell adhesion, migration and morphogenesis. Through a combination of experiments and mathematical modelling, the authors provide interesting insights into the roles of three non-muscle myosin isoforms in cellular morphodynamics and force generation.

**Decision letter after peer review:**

[Editors’ note: the authors submitted for reconsideration following the decision after peer review. What follows is the decision letter after the first round of review.]

Thank you for submitting your work entitled "Distinct roles of nonmuscle myosin II isoforms for establishing tension and elasticity during cell morphogenesis" for consideration by *eLife*. Your article has been reviewed by 2 peer reviewers, and the evaluation has been overseen by a Reviewing Editor and a Senior Editor. The following individual involved in review of your submission have agreed to reveal their identity: James R Sellers (Reviewer #1).

Our decision has been reached after consultation between the two reviewers and based on these discussions and the individual reviews below, we regret to inform you that your work will not be considered further for publication in *eLife*.

This paper is of interest for scientists studying cell migration, adhesion, and mechanosensing. The work provides interesting new information on the roles of non-muscle myosin II paralogs (NMIIA and NMIIB) in the mechanics of contractile actomyosin bundles. Through generating NMIIA, NMIIB and NMIIC knockout cell-lines, and analyzing their phenotypes on homogeneous and micropatterned substrates, the authors provide evidence that NMIIA is responsible for the generation of intracellular tension, whereas NMIIB elastically stabilizes the NMIIA-generated tension. They also performed fluorescence-recovery-after-photobleaching experiments, combined with mathematical modeling, to elucidate the role of different exchange kinetics of NMIIA and NMIIB in myosin minifilaments.

The data presented in the manuscript are of good technical quality. However, some conclusions presented in the manuscript are not particularly strongly supported by the experiments, and thus the study is somewhat preliminary at this stage. The detailed comments by the two reviewers can be found below. In the discussions among the reviewers it was also considered that, although the manuscript certainly provides new information on the different roles of NMIIA and NMIIB, this study does not present such fundamental new insight into the functions of NMII paralogs that would make it a strong candidate for publication in *eLife*.

*Reviewer #1:*

This manuscript examines the role of nonmuscle myosin IIA and IIB in establishing tension and elasticity in cells. They use CRISPR/Cas9 technology to ablate the specific paralogs and grow cells on micropatterned surfaces.

An unaddressed point, both experimentally and in the discussion and modeling, is what are the relative amounts of NM IIA and NM IIB in the U2OS cells? Also does the level of expression of one paralog change in response to ablation of the other?

On homogeneous substrates the authors show the localization of stress fibers and focal adhesions in both wild type and in the myosin paralog-specific KOs. They show the localizations of the myosin paralogs in the WT cells, but do not show the localizations of the remaining myosins in the paralog-specific KO cells. It would be informative to see this.

Line 190: I think there needs to be more discussion regarding the word "circular" as a description of arc shape.

Lines 195-197: Describe the origin of the values given for λ and σ.

NM IIA KO cells. The images shown in Figure 2—Figure supplement 1 does not appear to back up the statement that "only a few NM IIB minifilaments co-localize". It also makes me return to the first point made above about relative amounts of the myosins.

Modeling: The authors might want to use the term "duty ratio" to explain the difference in load bearing ability. Also, they do not mention the possibility of load-induced changes in the kinectis of these myosins. In this regard they should reference Kovacs et al. (2007)( doi: 10.1073/pnas.0701181104) which showed that the kinetics of both NM IIA and IIB are affected by load, but that NM IIB is more sensitive.

Lines 332-334. If the NM IIB filaments in the NM IIA KO cells are not phosphorylated, you might expect that blebbistatin might have little or no effect on these cells. Could this be tested?

Figure 3—figure supplement 2. I do not think these experiments add anything to the manuscript, unless more supporting information is provided. First of all, the details of this are not mentioned very prominently in the results. Second, the various mutants are used without any corroboration that they are actually behaving in the manner that is referenced. The authors should show (and quantify) the myosin filament localizations for these mutants.

Figure 5. Would you like to speculate on the "immobile" fraction of the myosin paralogs in the FRAP experiments? Do you envision that, perhaps, the myosin hexamers in the core of the filaments do not exchange? If so, that might not be consistent with your interpretations that these filaments can dissemble and form rapidly.

Line 388: There is actually quite a bit of controversy as to which kinetic step is correlated with force generation for myosins. Several studies suggested that force generation is associate with either ADP release or an isomerization of myosin-ADP states.

Blebbistatin inhibits the ability of myosin to enter a strongly bound state. In the presence of blebbistatin myosin binds only weakly to actin. If it inhibited the force-generating step then you would expect myosin to be strongly bound to actin in the presence of blebbistatin. Essentially, blebbistatin converts a phosphorylated, active NM II to a state that mimics unphosphorylated, inactive myosin.

Line 422. The Kovacs et al. (2007) paper mentioned above supports the notion that attachment lifetime of NM IIB is more force dependent than is that for NM IIA.

*Reviewer #2:*

This study investigates individual roles of three nonmuscle myosin II paralogs in U2OS cells using CRISPR/Cas9-mediated knockouts of each paralog. A novel aspect of this study is that the authors use cross-shaped fibronectin patterns to culture cells, which allowed them to evaluate quantitative aspects of the resulting phenotypes. As key metrics, they used a relationship [R(d)] between the curvature and the length of the lateral actin arcs formed on such patterns. The underlying hypothesis is that these parameters are defined by a competition between the surface tension all over the cell and the line tension in the arc. The authors also developed mathematical models to evaluate the ideas that the differences in both R(d) and the exchange rate of two myosins, NM IIA and NMIIB, result from distinct kinetics of their motors. These data reveal certain aspect about individual functions of NMII paralogs, although additional clarifications about underlying biology would be very helpful. Besides this main line of investigations, authors also present other observations, from which they draw conclusions, which are not sufficiently well justified.

1. The benefits of R(d) as the main parameter to characterize the differences in KO phenotypes are not obvious. Indeed, the difference between the IIA KO and IIB KO phenotypes is visually quite obvious, but it is not revealed by R(d). Can this parameter inform us about how the surface tension and/or the line tension changes in each case?

2. On the patterns, actin and myosin localize not only to arcs, where they apparently generate the line tension, but also to the cytoplasm over the passivated substrate, where they might generate surface tension. After IIB KO, more actin (Figure 2C) seems to move to the cytoplasm relative to how much remains in the arcs. Can it mean that these cells have higher surface tension and lower line tension relative to wild type? In this is the case, according to the proposed model, such redistribution should result in a higher curvature of the arcs, but the actual result was opposite – straighter arcs, which should mean that the line tension overwhelms surface tension. Does it mean then that IIB is mainly responsible for the surface tension? Is there a biological explanation for this result?

3. The assumption of a slower inflow from focal adhesions in the absence of IIA predicts straighter arcs. Conversely, a faster inflow in the absence of IIB should lead to more curved arcs. However, the results are opposite. Why do these intuitive considerations conflict with the conclusions of the model?

4. While interpreting the IIA KO phenotype, authors need to take into account that total amount of myosin II is significantly reduced in KO cells, as IIA is the major isoform in U2OS cells, suggesting that the phenotype could be well explained by a lower quantity, not by a different quality of the remaining myosin. A proper control would be to use cells that express IIB in the IIA KO cells at approximately the same level as total myosin II in WT cells.

5. The conclusion that IIA is necessary to initiate assembly of IIB filament is not supported by the data, which show that peripheral myosin II filaments in the cells have 75% of IIA subunits and 25% of IIB subunits (Figure 4-s2F), thus suggesting that all filaments initially contain both IIA and IIB, but their ratio changes over time and distance from the cell edge. No homotypic IIA filaments have been demonstrated in the study. Despite the conclusion saying "we found that all heterotypic minifilaments arise from homotypic NM IIA minifilaments" (p. 15, ll. 349-350). Available data in the literature show that IIB can polymerize by itself both in vitro and in cells lacking IIA. The claim that IIA or IIAΔIQ "restore" IIB filaments is not validated quantitatively. In fact, in figure 4-s1A, a NM IIA-KO cell that does not express GFP-NM IIA-WT has abundant NM IIB filaments. Moreover, the authors show that overexpression of IIB also restores NMIIB filaments (Figure 4-s1D), suggesting that a low levels of IIB is likely responsible for the low amount of IIB filaments in IIA KO cells, rather than their inability to form filaments in the absence of IIA.

6. The conclusions that IIA triggers RLC phosphorylation and that IIB can form filaments with unphosphorylated RLC are so extreme that their validation requires comprehensive analyses, extensive quantifications using proper normalizations to myosin levels, as well as alternative approaches. At the present state of knowledge, it is hard to imagine that myosin filaments would form without RLC phosphorylation. The idea that myosin II can somehow trigger a feedback loop to activate RLC phosphorylation is theoretically possible, but requires solid evidence, which is not provided here. The observations instigating the above conclusion are more likely explained by some technical issues. For example, IIB filaments may contain double-phosphorylated RLC, which is not recognized by the used antibody, or the amount of IIB is too low, or there is a problem with signal detection. The authors show that the pRLC level does increase linearly with overexpression of IIB although with a different slope compared with IIA. However, data in Figure 4-s1B and 4-s1E must come from different experiments, thus making pRLC staining intensities incomparable.

7. The significance of using the IIA mutants is hard to understand. First, it is not clear what mutants are meant in different statements, e.g. mutants with "prolonged NM IIA dwell times in the minifilaments" (p. 14, l. 313), or "mutants, in which the disassembly of the NM IIA hexamers was blocked" (p. 14, l. 315), or "constitutively active NMHC IIA construct" (p. 14, l. 317). They all seem to refer to mutants with impaired disassembly (ΔIQ2, ΔNHT and 3xA). Yet, they are contrasted to each other (p. 14, ll. 315-317 and in Figure 3-s2). Second, what is the idea behind using the ΔACD mutant? What does it reveal? Third, none of these mutations affects motor activity of IIA. They only affect its polymerization. Given that the mathematical model considers only motor activities of IIA and IIB, how do these experiments test the model? Finally, since IIB was not a part of these experiments, how did authors arrive to the following conclusions from these data: "This demonstrates that spatially and temporally balanced ratios of active NM IIA and NM IIB hexamers in heterotypic minifilaments are mandatory to adjust the contractile output in SFs and the relation between tension and elasticity. Therefore, the specific biochemical features of the isoforms and not their overall expression are important for the generation of tension and elastic stability, respectively." (p. 14, ll. 318-322) and "the specific intracellular force output is precisely tuned by the ratio and dwell time of individual NM IIA and NM IIB hexamers in the heterotypic minifilaments." (p.22, ll. 518-520)?

[Editors’ note: further revisions were suggested prior to acceptance, as described below.]

Thank you for submitting your article "Distinct roles of nonmuscle myosin II isoforms for establishing tension and elasticity during cell morphodynamics" for consideration by *eLife*. Your article has been reviewed by 2 peer reviewers, and the evaluation has been overseen by a Reviewing Editor and Anna Akhmanova as the Senior Editor. The following individual involved in review of your submission have agreed to reveal their identity: James R Sellers (Reviewer #1). Please note that because the original reviewer #2 was unable to evaluate the new submission, the manuscript was reviewed by another expert in the field.

Essential revisions:

While the reviewer #1 found the revised manuscript significantly improved, the reviewer #2 stated that large part of the data are confirmatory and that the most novel findings were not sufficiently well presented in the manuscript. Thus, the manuscript should be extensively rewritten to address the points raised by reviewer #2.

1). The study should be put better into a context of earlier work on NMII isoforms. The parts of the manuscript presenting confirmatory data should be shortened, and the most novel findings should be better explained to make them also accessible for a non-specialist reader. Making the manuscript shorter and more focused will increase its impact.

2). Also the 'Introduction' should be shortened and focused only on the published literature. Instead of extensively discussing new findings in the 'Introduction', these should be only briefly mentioned in the end of 'Introduction'.

*Reviewer #1:*

I am satisfied with the presentation of the data.

*Reviewer #2:*

This manuscript, previously revised in *eLife*, but not by this reviewer, describes the different effects of NMII isoforms in mechanical adaptation to different microenvironments. The approach consists of U2OS cells depleted of each specific isoform by CRISPR/CAS9. Based on the rebuttal, the authors had, in their previous version, data on NMIIA mutants as well as FRAP data, which have been removed from this iteration. Instead, the authors provide modeling to show that NMIIA is the "first responder" in generating tension; whereas NMIIB stabilizes elastic tension. The authors propose a novel role for NMIIC in establishing tensional homeostasis.

This manuscript contains important information regarding the role of NMII isoforms in cellular responses. The manuscript seems have changed mightily from its previous incarnation. Insomuch as this reviewer did not see the previous version, what follows is an appraisal on the current version.

Overall, the manuscript is well done, and experimentation is of high caliber. However, the study takes a long time getting into actually novel data, and its amount is limited. A significant part of the manuscript is confirmatory, including the role of NMIIA in force generation (Jorrisch et al., 2013, PMID 23616920, for example), adhesion elongation (many reports); and of NMIIB in adhesion "consolidation". The other reviewers asked about the relative amount of NMII isoforms, which was a good point. The authors have solved this by overexpressing NMIIB in NMIIA KO cells, which does not restore any effect observed in these cells, which actually confirms that the ability of NMIIB filaments to form is limited in these cells.

The authors engaged in an argument with the previous reviewers on the importance of the levels of NMII isoforms. While I'm convinced by the argument of the authors (NMIIB overexpression in NMIIA KOs is a good experiment), I'm curious as to more NMIIC has effects on the elastic recoil observed in the last experiment of the paper. Also, include mass spec data as in Ma et al. (2010) would be useful.

The novel part starts in figure 3, in which the authors observe a subtle change in the bending of actin bundles in cross-shaped patterns. The graph is quite counterintuitive. A and D look similar, and the graphs look similar. This is understandable. However, B and C (NMIIA and NMIIB Kos) are somewhat similar, yet the graphs are opposite, with dTEM converging on Rmin on NMIIA KOs; and away from Rmin in NMIIB KOs. The text explanation (pages 10 and 11) works, but the representative cell is head scratching.

I haven't seen the RLC phosphorylation data, but I'm intrigued. The manner the previous reviewers wrote about it makes it hard to understand what was going on. I'm guessing the authors will pursue this in future work.

In Figure 4, the authors propose a model that correlates dynamic tension and elasticity with actomyosin crossbridging. They propose that the short duty ratio of NMIIA correlates with the generation of dynamic tension; and the higher duty ratio of NMIIB explains the elastic behavior of the actomyosin arches. While it is entirely possible this may be true, the cellular behavior of myosin II chimeras (e.g. as published by Tony Means and Rick Horwitz) is not dominated by the duty ratio (which depends entirely on actin-myosin binding); but by myosin filamentation, which depends on the tail domains of the heavy chains. I would require the authors to integrate this in their model, which may be correct theoretically, but would hardly explain the behavior of the cell outside a cross-shaped pattern.

The most interesting argument is the potential role of NMIIC in the mechanical response of cells. The authors seem to consider NMIIC as an oscillatory dampener that controls force relaxation. However, this is a very undeveloped part of the manuscript, which merits further exploration. I don't think this is particularly easy.

In summary, while I find a lot of merit in this paper, I find that more than half the study is confirmatory, and the novel part will appeal only to hardcore specialists in the field. Thus, I am not convinced it represents a sufficient general advance for publication in *eLife*.

---

## [Author Response]

[Editors’ note: the authors resubmitted a revised version of the paper for consideration. What follows is the authors’ response to the first round of review.]

This paper is of interest for scientists studying cell migration, adhesion, and mechanosensing. The work provides interesting new information on the roles of non-muscle myosin II paralogs (NMIIA and NMIIB) in the mechanics of contractile actomyosin bundles. Through generating NMIIA, NMIIB and NMIIC knockout cell-lines, and analyzing their phenotypes on homogeneous and micropatterned substrates, the authors provide evidence that NMIIA is responsible for the generation of intracellular tension, whereas NMIIB elastically stabilizes the NMIIA-generated tension. They also performed fluorescence-recovery-after-photobleaching experiments, combined with mathematical modeling, to elucidate the role of different exchange kinetics of NMIIA and NMIIB in myosin minifilaments.The data presented in the manuscript are of good technical quality. However, some conclusions presented in the manuscript are not particularly strongly supported by the experiments, and thus the study is somewhat preliminary at this stage. The detailed comments by the two reviewers can be found below. In the discussions among the reviewers it was also considered that, although the manuscript certainly provides new information on the different roles of NMIIA and NMIIB, this study does not present such fundamental new insight into the functions of NMII paralogs that would make it a strong candidate for publication in eLife.

We thank the two editors and the two reviewers for their constructive and detailed comments, which motivated us to completely rework our project. We agree that our initial submission was somehow preliminary and now have added the results from new experiments that confirm our earlier statements on the cellular functions of the NM II paralogs. Strikingly, our new experiments even led to the identification of a novel cellular function for NM IIC. Our new advances became possible because we now complement our shape analysis by a force generation analysis:

– We have conducted new shape and force generation experiments for cells in 3D collagen gels and confirmed that NM IIA-KO cells can spread in a structured environment (as observed earlier for micropatterned substrates), but also that their invaginated arcs seem to be smaller, indicating smaller cortical tension. A gel contraction experiment then confirmed that they cannot generate global tension, in marked contrast to NM IIB-KO cells. In these experiments, no phenotype was observed for NM IIC-KO cells.

– In order to address the dynamics of force generation, we conducted new experiments in which single cells are dynamically stretched and relaxed in a 3D-printed stretching apparatus. As expected, we find that NM IIA-KO cells cannot generate forces in response to the changed environment. Strikingly, NM IIB-KO cells initially behaved similar to WT cells, but then generated significantly less force upon a mechanical stretch. Very surprisingly, however, we now found for the first time a phenotype for NM IIC-KO cells, namely the inability to establish tensional homeostasis after stretch release.

In response to the comments by the reviewers, we now have also conducted NM IIB overexpression experiments that demonstrate that although NM IIA is much more abundant than NM IIB, their different functions as reported by us is qualitative in nature and does not depend on the exact expression level. Because our new experiments now increase the focus on the cellular function of the different NM II paralogs, we decided to leave out our preliminary results on RLC-phosphorylation and exchange dynamics, which we will improve and publish in future work.

In summary, we have completely reworked this manuscript by performing several new types of experiments. We believe that they have strengthened the validity of our earlier results on the distinct functions of NM IIA and B by clarifying the role of expression levels, physiological 3D environments and force generation. In addition, we have discovered a new and formerly unknown cellular function of NM IIC. Together, these results make our revised manuscript much very interesting to the general readers of *eLife* and we hope that you will be able to reconsider our revised manuscript for publication.

Reviewer #1:This manuscript examines the role of nonmuscle myosin IIA and IIB in establishing tension and elasticity in cells. They use CRISPR/Cas9 technology to ablate the specific paralogs and grow cells on micropatterned surfaces.An unaddressed point, both experimentally and in the discussion and modeling, is what are the relative amounts of NM IIA and NM IIB in the U2OS cells? Also does the level of expression of one paralog change in response to ablation of the other?

We thank the reviewer for these questions, which we were able to answer in additional experiments. We have generated GFP-NMHC IIA and GFP-NMHC IIB fusion proteins with the GFP expressed under the endogenous promoter of the paralog. This enabled us to precisely quantify the relative protein amounts of NM IIA and NM IIB without overexpression or antibody binding artifacts. We added the quantification in the new Figure 2B to clarify this aspect. We also measured the change of expression of the paralogs in response to the ablation of the other, as suggested by the reviewer and added this info to the new Figure 1—figure supplement 3E. As expected by the reviewer, we find that NM IIA is much more abundant than NM IIB (in fact 4.5 times more abundant), but that this abundance does not change the qualitative difference between the two paralog functions, namely that NM IIA is required to establish tension and that NM IIB stabilizes it.

On homogeneous substrates the authors show the localization of stress fibers and focal adhesions in both wild type and in the myosin paralog-specific KOs. They show the localizations of the myosin paralogs in the WT cells, but do not show the localizations of the remaining myosins in the paralog-specific KO cells. It would be informative to see this.

The localization of the remaining paralog (for NM IIA and NM IIB) had been shown in Figure 4 of the initial submission. Since we have now included new experiments in our revised manuscript, we not only have substantially rewritten the text, but also rearranged the figures accordingly. We removed old Figure 4 and instead provide a more detailed analysis about the localization of the remaining paralogs in new Figure 1—figure supplement 3. These data now provide the information requested by the reviewer. In summary, we find that the depletion of NM IIA does not significantly increase the amount of NM IIB or NM IIC minifilaments along SFs. However, the loss of NM IIA leads to a changed localization pattern of NM IIB and NM IIC. Since such changes in paralog localization were not observed for NM IIB-KO and NM IIC-KO cells, these results again strengthen our interpretation, namely that NM IIA initiates new contraction sites and guides the actomyosin system in the absence of external guidance cues.

Line 190: I think there needs to be more discussion regarding the word "circular" as a description of arc shape.

We now provide a quantification of circularity as new Figure 3—figure supplement 2. First the cell contours were fit with the ImageJ/Fiji-plugin JFilament and then circular arcs very fitted using a custom-written script. Next the root mean squared deviation between points on the contour and the fitted circle was measured and converted into a relative error statement. We find that for each cell type the average error was always below 2 percent, demonstrating that these arcs are indeed strongly circular, as observed first in our earlier paper Bischofs et al. Biophysical Journal 2008 (10.1529/biophysj.108.134296) and later confirmed by many other groups. The error was largest for the NM IIA-KOs, in agreement with our main result that these cells cannot generate tension.

Lines 195-197: Describe the origin of the values given for λ and σ.

This statement is based on model fits that have been performed by many groups with roughly similar results. We now cite some of the corresponding papers for these values. The basic idea is that the radius of the circular arc should follow the Laplace law R=λ/σ. Because R can be easily measured by circle fitting, one still has to estimate either λ or σ from another process. This can be done e.g. with traction force microscopy on soft elastic substrates or on pillar arrays, compare our paper Bischofs, Schmidt and Schwarz, Physical Reviews Letters 2009 (10.1103/PhysRevLett.103.048101), which we now cite in the revised version. In the current work, we do not provide explicit estimates for λ or σ and only the order of magnitude values are given as in the initial submission. It is important to understand that NM II KO will change both λ and σ and therefore we cannot predict changes on radius R=λ/σ. From our earlier work on cell shapes (10.1016/j.celrep.2019.04.035; 10.1529/biophysj.108.134296), we know that blebbistatin inhibition of NM II tends to affect λ more than σ, leading to smaller radii. As we show here, the same reasoning cannot be applied directly to the NM II-KOs, because besides force, also flow seems to play an important role (NM IIB has a larger stall force, but also a higher effective friction, and these effect work against each other in our model). We leave it to future work to directly measure exact values for λ or σ for the NM II-KOs investigated here. Our main focus here is to show that in principle, the experimentally measured R-d-relations can be explained directly from the known differences in the NM IIA and B crossbridge cycles.

NM IIA KO cells. The images shown in Figure 2—Figure supplement 1 does not appear to back up the statement that "only a few NM IIB minifilaments co-localize". It also makes me return to the first point made above about relative amounts of the myosins.

As stated above, we have improved this part by including measurements about the paralog intensities. Although we did not find a significantly decreased intensity of the NM IIB filaments when NM IIA was depleted, their localization was changed. The NM IIB filaments in NM IIA-KO cells are, however, so densely packed that we were not able to extract the number of filaments along the peripheral actin arcs. We believe that this issue is now clarified by the other changes.

Modeling: The authors might want to use the term "duty ratio" to explain the difference in load bearing ability. Also, they do not mention the possibility of load-induced changes in the kinectis of these myosins. In this regard they should reference Kovacs et al. (2007)( doi: 10.1073/pnas.0701181104) which showed that the kinetics of both NM IIA and IIB are affected by load, but that NM IIB is more sensitive.

Thank you for this suggestion. We now mentioned in the main text that NM IIA has a much smaller duty ratio than NM IIB. We note that our model in principle includes the load dependance of the different myosins (compare theory supplement), but that our simulations show that the faster release rate of NM IIA is the dominating aspect. In our model, the load-dependent rates mainly affect the mechanics of the motor in such a way that a higher affinity to actin at higher loads leads to increases in the convexity of the force-velocity relation and the stall force. This effect is negligible for the low duty ratio of NM IIA, while it becomes appreciable for NM IIB (see the theory supplement Figure S1C). Reducing the catch-bond force scale f_c_ for NM IIB (i.e. increasing the affinity to actin at high forces further) only strengthens this behavior. We now included this discussion in the theory supplement.

Lines 332-334. If the NM IIB filaments in the NM IIA KO cells are not phosphorylated, you might expect that blebbistatin might have little or no effect on these cells. Could this be tested?

It was never our intention to create the impression that NM IIB filaments are not phosphorylated in general and we apologize if our statements and conclusions were misleading in this context. Our initial speculation was that NM IIA induces a feedback loop which might enhance the phosphorylation of NM IIB. We agree with the reviewer that this could be tested by using blebbistatin. However, because we now have shifted the focus of this work to cellular effects, we decided to not cover this interesting question here, but to investigate it further in future work.

Figure 3—figure supplement 2. I do not think these experiments add anything to the manuscript, unless more supporting information is provided. First of all, the details of this are not mentioned very prominently in the results. Second, the various mutants are used without any corroboration that they are actually behaving in the manner that is referenced. The authors should show (and quantify) the myosin filament localizations for these mutants.

We thank the reviewer for pointing this out. Indeed, the description of this part was very condensed due to space limitations. We have now decided to remove this figure supplement as part of our effort to improve the manuscript’s focus towards cell mechanical issues.

Figure 5. Would you like to speculate on the "immobile" fraction of the myosin paralogs in the FRAP experiments? Do you envision that, perhaps, the myosin hexamers in the core of the filaments do not exchange? If so, that might not be consistent with your interpretations that these filaments can dissemble and form rapidly.

This is a very interesting suggestion that should be addressed with super-resolution approaches in the future. Even if there was a core that did not disassemble, that smaller assembly could more easily rearrange and reassemble. We now have removed the FRAP experiments from our manuscript to improve the focus on cell mechanics, but will address this important issue in future work.

Line 388: There is actually quite a bit of controversy as to which kinetic step is correlated with force generation for myosins. Several studies suggested that force generation is associate with either ADP release or an isomerization of myosin-ADP states.Blebbistatin inhibits the ability of myosin to enter a strongly bound state. In the presence of blebbistatin myosin binds only weakly to actin. If it inhibited the force-generating step then you would expect myosin to be strongly bound to actin in the presence of blebbistatin. Essentially, blebbistatin converts a phosphorylated, active NM II to a state that mimics unphosphorylated, inactive myosin.

Many thanks for these explanations. We agree that blebbistatin inhibits the transition to the strongly bound state and in our model this is represented by reducing the rate k_12_. As we now have removed the FRAP-part of our manuscript, this part is no longer needed, but will be used in a future publication. However, in the theory supplement we briefly comment on the non-stereospecific binding of myosin II to actin discovered in the context of blebbistatin inhibition and cite the corresponding paper by the Mansson group.

Line 422. The Kovacs et al. (2007) paper mentioned above supports the notion that attachment lifetime of NM IIB is more force dependent than is that for NM IIA.

Many thanks for this comment. In our model, the main effect of A versus B is the fast rate for dissociation from the filament. This step is force-dependent as explained in the supplement, but we do not assign different force scales to the two isoforms. However, there is an indirect effect in our model consistent with the Kovacs-paper, namely that the slowly cycling B is stronger force-dependent because it spends more time on the filament, and therefore the powerstroke becomes more important, although here we take it not to be explicitly force-dependent. In principle, we could further adapt our model according to this suggestion by also assigning different force scales to A versus B. This might in fact be an important aspect when addressing different loading situations, but in general, we take it from our simulations that the dominating effect will be the differences in dissociation rates as already implemented and sufficient for our main conclusions. In the absence of more quantitative data, we prefer to keep the simple model.

Reviewer #2:This study investigates individual roles of three nonmuscle myosin II paralogs in U2OS cells using CRISPR/Cas9-mediated knockouts of each paralog. A novel aspect of this study is that the authors use cross-shaped fibronectin patterns to culture cells, which allowed them to evaluate quantitative aspects of the resulting phenotypes. As key metrics, they used a relationship [R(d)] between the curvature and the length of the lateral actin arcs formed on such patterns. The underlying hypothesis is that these parameters are defined by a competition between the surface tension all over the cell and the line tension in the arc. The authors also developed mathematical models to evaluate the ideas that the differences in both R(d) and the exchange rate of two myosins, NM IIA and NMIIB, result from distinct kinetics of their motors. These data reveal certain aspect about individual functions of NMII paralogs, although additional clarifications about underlying biology would be very helpful. Besides this main line of investigations, authors also present other observations, from which they draw conclusions, which are not sufficiently well justified.

We thank the reviewer for this candid assessment and believe that these issues are all addressed in the revised version. Please note that we now have added several types of additional experiments (overexpression experiments, cell shape and force generation in 3D collagen gels, dynamics of force generation in 3D scaffolds) that confirm and deepen our conclusions obtained with the micropattern experiments.

1. The benefits of R(d) as the main parameter to characterize the differences in KO phenotypes are not obvious. Indeed, the difference between the IIA KO and IIB KO phenotypes is visually quite obvious, but it is not revealed by R(d). Can this parameter inform us about how the surface tension and/or the line tension changes in each case?

We first note that the phenotypes on homogeneous and structured substrates are complementary and that this is a major and very important result of our work. While the observations on the homogeneous substrates seem to suggest that NM IIA-KO cells cannot establish a contracted and structured phenotype, the micropatterns demonstrate that in a structured environment, these cells do establish a distinct phenotype and build up some level of tension, because also here they form invaginated arcs. The observations on the homogeneous substrates seem to suggest that NM IIBKO cells are rather similar to WT, but the micropatterns demonstrate that the R-drelation breaks down, revealing that NM IIB is required for stabilization. We conclude that the phenotypes on homogenous and structured substrates complement each other and irrespective of the exact values extracted for σ and λ, the micropattern assay reveals important functions of the different paralogs that cannot be deduced directly on the homogenous substrates. Note that with the cell stretching experiments, we now also have found a way to identify a cellular function of NM IIC. In addition, we now have added the collagen experiments to demonstrate that the conclusions drawn from the micropatterns also manifest themselves in the more physiological context of a tissue environment.

Regarding the exact values of the two parameters, the Laplace law R=λ/σ can be used only to extract the ratio of the two, thus one needs additional experiments to determine absolute values. In our previous paper Bischofs, Schmidt and Schwarz Physical Review Letters 2009 (10.1103/PhysRevLett.103.048101), we showed how this could be done with traction forces on pillar arrays. This line of research, however, is not the focus of the work presented here, where we aimed at explaining the R-d relation directly from the known differences between the crossbridge cycle of A versus B without having to specify values for σ and λ.

In order to make progress towards a better understanding of cell mechanics as a function of NM II-KO, we now have performed additional nanoindentation experiments using AFM force spectroscopy to confirm changes in σ after KO (shown in Figure 2F of our new manuscript). We found that on the dorsal side above the nuclear region, where only surface tension should contribute to the actomyosin contractility, NM IIAKO cells are significantly softer than WT cells, while NM IIB-KO cells are significantly stiffer. Because these experiments measure an effective Young’s modulus by fitting to a Hertz indentation curve, we cannot extract specific values for σ, but it is clear from these experiments that σ is strongly reduced in the NM IIA-KO cells because without NM IIA, the whole force generating machinery does not work properly. This is also confirmed by the cell stretching experiments, in which NM IIA-KO could not generate any force.

2. On the patterns, actin and myosin localize not only to arcs, where they apparently generate the line tension, but also to the cytoplasm over the passivated substrate, where they might generate surface tension. After IIB KO, more actin (Figure 2C) seems to move to the cytoplasm relative to how much remains in the arcs. Can it mean that these cells have higher surface tension and lower line tension relative to wild type? In this is the case, according to the proposed model, such redistribution should result in a higher curvature of the arcs, but the actual result was opposite – straighter arcs, which should mean that the line tension overwhelms surface tension. Does it mean then that IIB is mainly responsible for the surface tension? Is there a biological explanation for this result?

We agree with the reviewer that one might get the impression from the mentioned figure that more actin is translocated to the cytoplasm over the passivated area and we therefore followed up on this interesting comment. When quantifying the amount of actin fibers by averaging the coherency of the structure tensor as an indirect measure of fiber density, we found a trend towards more actin fibers over the passive area in case of NM IIB-KO cells. However, the difference was not significant between WT and NM IIB-KO cells (now shown in Figure 3—figure supplement 3A) and we want to emphasize that such high actin densities are not abundant in each and every NM IIBKO cell, thus reflecting the high heterogeneity of the cells, as also stated in the manuscript.

The question regarding the surface tension and line tension in NM IIB-KO cells is an interesting point that is, however, not easy to answer. First, we want to point out that the internal actin fibers above the passivated area resemble ventral stress fibers, which in this case should not contribute to the surface tension, since they are attached to the substrate at both ends on the ventral side of the cell (see Figure 3). However, the assumption that the cortical (surface) tension should also be influenced by the loss of NM IIB is plausible and an increase in cortex tension was recently postulated in Taneja et al. 2020 (https://doi.org/10.1016/j.celrep.2020.03.041). Since this paper, however, dealt with the process of cytokinesis, the cells were in suspension during the analysis, therefore lacking any stress fibers. In our set-up, both, stress fibers (line tension) and cortex (surface tension) contribute to force generation and we believe that both are driven by a global pool of NM II molecules that generate the force output. This is supported by our new data from the AFM nanoindentation experiments, where a higher surface tension was measured in NM IIB-KO cells and a lower surface tension in NM IIA-KO cells. Although we cannot deduce the proportions of NM II molecules contributing to surface or line tension from these experiments, these data suggest that a loss of NM IIB leads to both, higher surface tension and higher line tension, while it is the opposite for NM IIA-KO cells, where both tensions are reduced.

Finally, we want to highlight that a differential distribution of the different NM II paralogs to surface tension or line tension might still be possible and it would be a very interesting phenomenon to tackle. We refrain from giving explicit values for λ and σ in this study and leave this important subject to future work.

3. The assumption of a slower inflow from focal adhesions in the absence of IIA predicts straighter arcs. Conversely, a faster inflow in the absence of IIB should lead to more curved arcs. However, the results are opposite. Why do these intuitive considerations conflict with the conclusions of the model?

We thank the reviewer for this interesting question. Our theory shows that the situation is more complex and requires to consider both flows and forces. Put simply, however, one can argue in the following manner. For NM IIA-KO, when only NM IIB is present, the flow is slower (larger friction) and the focal adhesions give out more length (at constant force) than can be accommodated by the slowly flowing fiber. Therefore the fiber has to grow longer and its length increases, leading to smaller radii, as observed experimentally. Note that this reasoning does not replace the full theory as given in the supplemental.

4. While interpreting the IIA KO phenotype, authors need to take into account that total amount of myosin II is significantly reduced in KO cells, as IIA is the major isoform in U2OS cells, suggesting that the phenotype could be well explained by a lower quantity, not by a different quality of the remaining myosin. A proper control would be to use cells that express IIB in the IIA KO cells at approximately the same level as total myosin II in WT cells.

Many thanks for raising this important point, which was also brought up by the other reviewer. As pointed out in the response to the comments of reviewer 1, we measured ratios of NM IIA to NM IIB and found that the ratio of NM IIA to NM IIB along SFs is approx. 4.5 to 1. To address the concerns of the reviewers that the total amount of NM II molecules rather than the different quality of the isoforms is the reason for the different phenotypes, we did some additional experiments, including the suggested control. We measured the amount of NM IIA in WT cells that express GFP under the endogenous NM IIA promotor and subsequently overexpressed NM IIB-GFP in NM IIA-KO cells under a constitutively active promotor to reach maximal expression of NM IIB. We measured NM IIB intensities along SFs that were 1.7 fold higher than the intensity of GFP-NM IIA in WT cells. Additionally, NM IIA-KO cells express endogenous NM IIB, which was not taken into account in these measurements. Thus, the real number of NM IIB molecules was even higher. However, even under these conditions, NM IIB was not able to phenocopy the NM II filament distribution, the pRLC intensity, the SF formation, the FA size or the morphology of WT cells. Moreover, overexpression of NM IIB did also not affect the R(d)-Relation on the micropattern (Figure 3—figure supplement 4). Finally, as already stated above, our AFM nanoindentation experiments show that the loss of NM IIA and NM IIB lead to opposite effects, namely a loss of surface tension in NM IIA-KO cells and an increase of surface tension in NM IIB-KO cells. These results are not consistent with the idea that reduced quantities of NM II molecules lead to the observed phenotypes, since in this scenario one would expect similar effects of different amplitudes rather than opposite outcomes. Our new results are now shown in Figure 2.

5. The conclusion that IIA is necessary to initiate assembly of IIB filament is not supported by the data, which show that peripheral myosin II filaments in the cells have 75% of IIA subunits and 25% of IIB subunits (Figure 4-s2F), thus suggesting that all filaments initially contain both IIA and IIB, but their ratio changes over time and distance from the cell edge. No homotypic IIA filaments have been demonstrated in the study. Despite the conclusion saying "we found that all heterotypic minifilaments arise from homotypic NM IIA minifilaments" (p. 15, ll. 349-350). Available data in the literature show that IIB can polymerize by itself both in vitro and in cells lacking IIA. The claim that IIA or IIAΔIQ "restore" IIB filaments is not validated quantitatively. In fact, in figure 4-s1A, a NM IIA-KO cell that does not express GFP-NM IIA-WT has abundant NM IIB filaments. Moreover, the authors show that overexpression of IIB also restores NMIIB filaments (Figure 4-s1D), suggesting that a low levels of IIB is likely responsible for the low amount of IIB filaments in IIA KO cells, rather than their inability to form filaments in the absence of IIA.

The reviewer is correct in stating that NM IIB filaments can also assemble on their own and this also happens in our cells as shown in the new Figure 1—figure supplement 3 and Figure 3—figure supplement 3. We apologize if there existed misleading statements in this regard. Our point is, however, that NM IIB filaments seemingly arise in much lower numbers when NM IIA is not present and as suggested by the reviewer itself in the next bullet point, we believe that NM IIA can somehow trigger a feedback loop to enhance RLC phosphorylation and NM IIB filament distribution. However, we also agree that this hypothesis requires solid evidence, which was not provided in our initial manuscript. We therefore decided to follow up on this very interesting biochemical topic in future work and to shift the scope of this study towards the cellular effects.

6. The conclusions that IIA triggers RLC phosphorylation and that IIB can form filaments with unphosphorylated RLC are so extreme that their validation requires comprehensive analyses, extensive quantifications using proper normalizations to myosin levels, as well as alternative approaches. At the present state of knowledge, it is hard to imagine that myosin filaments would form without RLC phosphorylation. The idea that myosin II can somehow trigger a feedback loop to activate RLC phosphorylation is theoretically possible, but requires solid evidence, which is not provided here. The observations instigating the above conclusion are more likely explained by some technical issues. For example, IIB filaments may contain double-phosphorylated RLC, which is not recognized by the used antibody, or the amount of IIB is too low, or there is a problem with signal detection. The authors show that the pRLC level does increase linearly with overexpression of IIB although with a different slope compared with IIA. However, data in Figure 4-s1B and 4-s1E must come from different experiments, thus making pRLC staining intensities incomparable.

We again apologize that we have not clearly communicated our statement. We do not believe that NM IIB filaments form without RLC phosphorylation. As pointed out above, our data point towards the direction that there might be a positive feedback loop (as also suggested by the reviewer) that is initiated by NM IIA rather than NM IIB. However, to entangle the complete molecular mechanism underlying such a system is beyond the scope of this study. We still believe that these data provide interesting new insights that do not arise from technical issues and that might help to entangle such mechanisms in the future. Yet we removed the according data to publish it in the future, when all necessary controls are done. This allowed us to focus more on the mechanistic roles of the isoforms and we included the new data regarding the force generation and reactive forces upon mechanical stress in 3D environments. As already stated above, we thus believe that due to these adjustments, our manuscript has significantly improved in both, distinctness and relevance.

7. The significance of using the IIA mutants is hard to understand. First, it is not clear what mutants are meant in different statements, e.g. mutants with "prolonged NM IIA dwell times in the minifilaments" (p. 14, l. 313), or "mutants, in which the disassembly of the NM IIA hexamers was blocked" (p. 14, l. 315), or "constitutively active NMHC IIA construct" (p. 14, l. 317). They all seem to refer to mutants with impaired disassembly (ΔIQ2, ΔNHT and 3xA). Yet, they are contrasted to each other (p. 14, ll. 315-317 and in Figure 3-s2). Second, what is the idea behind using the ΔACD mutant? What does it reveal? Third, none of these mutations affects motor activity of IIA. They only affect its polymerization. Given that the mathematical model considers only motor activities of IIA and IIB, how do these experiments test the model? Finally, since IIB was not a part of these experiments, how did authors arrive to the following conclusions from these data: "This demonstrates that spatially and temporally balanced ratios of active NM IIA and NM IIB hexamers in heterotypic minifilaments are mandatory to adjust the contractile output in SFs and the relation between tension and elasticity. Therefore, the specific biochemical features of the isoforms and not their overall expression are important for the generation of tension and elastic stability, respectively." (p. 14, ll. 318-322) and "the specific intracellular force output is precisely tuned by the ratio and dwell time of individual NM IIA and NM IIB hexamers in the heterotypic minifilaments." (p.22, ll. 518-520)?

We agree with the reviewer regarding the description of the mutants and apologize for not stating this more clearly in the manuscript due to space limitations. We now understand that we have tried to pack too many results into our manuscript and we have now removed the supplemental figure and explanations regarding it completely to improve the focus of the paper towards cell mechanical phenomena.

[Editors’ note: what follows is the authors’ response to the second round of review.]

Essential revisions:While the reviewer #1 found the revised manuscript significantly improved, the reviewer #2 stated that large part of the data are confirmatory and that the most novel findings were not sufficiently well presented in the manuscript. Thus, the manuscript should be extensively rewritten to address the points raised by reviewer #2.1). The study should be put better into a context of earlier work on NMII isoforms. The parts of the manuscript presenting confirmatory data should be shortened, and the most novel findings should be better explained to make them also accessible for a non-specialist reader. Making the manuscript shorter and more focused will increase its impact.2). Also the 'Introduction' should be shortened and focused only on the published literature. Instead of extensively discussing new findings in the 'Introduction', these should be only briefly mentioned in the end of 'Introduction'.

We agree that our revised version was unnecessarily detailed and contained many explanations and repetitions that were not needed for a concise presentation. We now have generated a significantly shortened version with a strong focus on the newly achieved results. In particular, we emphasize already in the abstract that the novel aspect is the quantitative analysis of the cells in a structured environment, and how this allows us to extract statements on the different dynamical roles of the different NM II isoforms. The confirmatory parts in the main text have been drastically shortened to increase the focus of the work and the novel findings are highlighted to make them more accessible for non-specialized readers. The Introduction has also been heavily shortened and now focuses on the necessary background literature. All carried out changes are highlighted in the DOCX text file using the track changes function with the following color code: Removed parts are marked in red, novel parts in green, and adjusted parts are in blue.

Reviewer #2:This manuscript, previously revised in eLife, but not by this reviewer, describes the different effects of NMII isoforms in mechanical adaptation to different microenvironments. The approach consists of U2OS cells depleted of each specific isoform by CRISPR/CAS9. Based on the rebuttal, the authors had, in their previous version, data on NMIIA mutants as well as FRAP data, which have been removed from this iteration. Instead, the authors provide modeling to show that NMIIA is the "first responder" in generating tension; whereas NMIIB stabilizes elastic tension. The authors propose a novel role for NMIIC in establishing tensional homeostasis.This manuscript contains important information regarding the role of NMII isoforms in cellular responses. The manuscript seems have changed mightily from its previous incarnation. Insomuch as this reviewer did not see the previous version, what follows is an appraisal on the current version.Overall, the manuscript is well done, and experimentation is of high caliber. However, the study takes a long time getting into actually novel data, and its amount is limited. A significant part of the manuscript is confirmatory, including the role of NMIIA in force generation (Jorrisch et al., 2013, PMID 23616920, for example), adhesion elongation (many reports); and of NMIIB in adhesion "consolidation". The other reviewers asked about the relative amount of NMII isoforms, which was a good point. The authors have solved this by overexpressing NMIIB in NMIIA KO cells, which does not restore any effect observed in these cells, which actually confirms that the ability of NMIIB filaments to form is limited in these cells.

Although similar results have been described before using RNA-interference and genetic ablation in mice, this is to our knowledge the first time that CRISPR/Cas9-based depletions of all three isoforms were generated from the same cellular background and analysed in such a comparable manner. After intense discussions, we are convinced that it is necessary to show the analysis for homogeneous substrates (Figure 1) because these data provide a basis for our main results. In the revised manuscript, we have drastically shortened the confirmatory text sections and now clearly state that these results are not shown because of their novelty, but for the sake of comparison. In principle, it would also be possible to move Figure 2 with the overexpression data into the supplement, but this part of the revised manuscript was earlier suggested by both reviewers and also appreciated now by the new reviewer. Because we now have strongly shortened the main text, we believe that the figures can remain as they are, but if advised otherwise, we are happy to move Figure 2 into the supplement.

The authors engaged in an argument with the previous reviewers on the importance of the levels of NMII isoforms. While I'm convinced by the argument of the authors (NMIIB overexpression in NMIIA KOs is a good experiment), I'm curious as to more NMIIC has effects on the elastic recoil observed in the last experiment of the paper. Also, include mass spec data as in Ma et al. (2010) would be useful.

We agree with the reviewer that such data would be very interesting. However, as also pointed out correctly by the reviewer his last comment, these experiments require extensive amounts of additional work and experiments. To systematically and quantitatively analyze NM IIC overexpression in our cell stretching assay, the cells should stably overexpress NM IIC, which would at least take three months. Following the editorial advice, we now focus on producing a concise version of our manuscript, which will be more accessible for the general reader of *eLife*.

The novel part starts in figure 3, in which the authors observe a subtle change in the bending of actin bundles in cross-shaped patterns. The graph is quite counterintuitive. A and D look similar, and the graphs look similar. This is understandable. However, B and C (NMIIA and NMIIB Kos) are somewhat similar, yet the graphs are opposite, with dTEM converging on Rmin on NMIIA KOs; and away from Rmin in NMIIB KOs. The text explanation (pages 10 and 11) works, but the representative cell is head scratching.

We agree with the reviewer that it is difficult to select representative images for a statistical analysis. After intense discussion, we are convinced that our choices for NM IIA-KO and NM IIB-KO cells make sense, because they reflect two of our major findings: First, the phenotype of NM IIA-KO cells is dampened on the cross-shaped micropattern and more closely resembles the WT phenotype than on homogenous substrates. Second, although the phenotype of NM IIB-KO cells is comparable to the WT situation by visual inspection, the quantification shows a marked difference, namely the loss of R(d) correlation, which arises from the mixed population of bent and almost straight actin arcs. In Figure 3C, we have chosen a cell that in our opinion reflects this heterogeneity, since it shows two almost straight and two more bent arcs. However, we again want to highlight that this striking result can only be revealed by the statistical analysis. We have added some more clarifying statements in the text, making this more transparent.

I haven't seen the RLC phosphorylation data, but I'm intrigued. The manner the previous reviewers wrote about it makes it hard to understand what was going on. I'm guessing the authors will pursue this in future work.

We are happy that the reviewer was already intrigued regarding the response to our pRLC data, as we are also excited and wish to present these data soon in a follow-up publication. Since the data were deemed too preliminary by the former reviewer(s), we removed these data to increase the focus of our revised manuscript.

In Figure 4, the authors propose a model that correlates dynamic tension and elasticity with actomyosin crossbridging. They propose that the short duty ratio of NMIIA correlates with the generation of dynamic tension; and the higher duty ratio of NMIIB explains the elastic behavior of the actomyosin arches. While it is entirely possible this may be true, the cellular behavior of myosin II chimeras (e.g. as published by Tony Means and Rick Horwitz) is not dominated by the duty ratio (which depends entirely on actin-myosin binding); but by myosin filamentation, which depends on the tail domains of the heavy chains. I would require the authors to integrate this in their model, which may be correct theoretically, but would hardly explain the behavior of the cell outside a cross-shaped pattern.

We completely agree that our explanations only concern the heads and not the tails. As now explained in more detail in the manuscript, there are two reasons for this approach. First, the focus on force generation rather than minifilament polymerization is a logical consequence of our earlier work to explain the R(d)-relation by force generation in bundles (line tension λ) and networks (surface tension σ). Second, the differences in the crossbridge cycles are sufficient to explain our data observed on the micropatterns. This being said, we share the concerns of the reviewer and plan to extend our experimental and modelling approaches in this direction in the future. This is now being stated in the discussion, where we also write that experiments with chimeras would be the way to go forward.

The most interesting argument is the potential role of NMIIC in the mechanical response of cells. The authors seem to consider NMIIC as an oscillatory dampener that controls force relaxation. However, this is a very undeveloped part of the manuscript, which merits further exploration. I don't think this is particularly easy.

We completely agree. These results have been found in response to the comments of the first revision and are very exciting, but need further scrutiny in future work, e.g. generation of cell line stably overexpressing NM IIC. However, as the reviewer again pointed out correctly, this will not be particularly easy but requires some in-depth analysis with advanced techniques. Given the focus of the current manuscript, we make a very clear statement that certainly is of large interest for the readers of *eLife*: NM IIC has no measurable role in cell shape determination, but in tensional homeostasis, which is a more dynamic process. We now emphasize this aspect in more detail in the discussion.

In summary, while I find a lot of merit in this paper, I find that more than half the study is confirmatory, and the novel part will appeal only to hardcore specialists in the field. Thus, I am not convinced it represents a sufficient general advance for publication in eLife.

We believe that our revised manuscript has significantly increased its focus and accessibility. We hope that in its new version it is now suited for publication in *eLife*.